# Commensal microflora-induced T cell responses mediate progressive neurodegeneration in glaucoma

Huihui Chen[1,2,3,4], Kin-Sang Cho [3,4], T. H. Khanh Vu[3,4,5], Ching-Hung Shen[6,7], Mandeep Kaur[6,7], Guochun Chen[3,4,8], Rose Mathew[3], M. Lisa McHam[9], Ahad Fazelat[3], Kameran Lashkari[3,4], Ngan Pan Bennett Au[10], Joyce Ka Yu Tse[10], Yingqian Li[3,4], Honghua Yu[3,4], Lanbo Yang[3], Joan Stein-Streilein[3,4], Chi Him Eddie Ma[10,11,12], Clifford J. Woolf [11,12], Mark T. Whary[13,14], Martine J. Jager [5], James G. Fox[13,14], Jianzhu Chen[6,7] & Dong F. Chen [3,4]

Glaucoma is the most prevalent neurodegenerative disease and a leading cause of blindness worldwide. The mechanisms causing glaucomatous neurodegeneration are not fully understood. Here we show, using mice deficient in T and/or B cells and adoptive cell transfer, that transient elevation of intraocular pressure (IOP) is sufficient to induce T-cell infiltration into the retina. This T-cell infiltration leads to a prolonged phase of retinal ganglion cell degeneration that persists after IOP returns to a normal level. Heat shock proteins (HSP) are identified as target antigens of T-cell responses in glaucomatous mice and human glaucoma patients. Furthermore, retina-infiltrating T cells cross-react with human and bacterial HSPs; mice raised in the absence of commensal microflora do not develop glaucomatous T-cell responses or the associated neurodegeneration. These results provide compelling evidence that glaucomatous neurodegeneration is mediated in part by T cells that are pre-sensitized by exposure to commensal microflora.

[1] Department of Ophthalmology, Second Xiangya Hospital of Central South University Changsha, Hunan Province, 410011 Hunan, China. [2] Institution of Ophthalmic Center, Changsha, Hunan Province, 410011 Hunan, China. [3] Schepens Eye Research Institute of Massachusetts Eye and Ear, Harvard Medical School, Boston, 02114 MA, USA. [4] Department of Ophthalmology, Harvard Medical School, Boston, 02114 MA, USA. [5] Department of Ophthalmology, Leiden University Medical Center, 2333 ZA, Leiden, The Netherlands. [6] Koch Institute for Integrative Cancer Research, Massachusetts Institute of Technology, Cambridge, 02139 MA, USA. [7] Department of Biology, Massachusetts Institute of Technology, Cambridge, 02139 MA, USA. [8] Department of Nephrology, Second Xiangya Hospital of Central South University, Changsha, 410011 Hunan, China. [9] Massachusetts Eye Health Service, Boston, 02124 MA, USA. [10] Department of Biomedical Sciences, City University of Hong Kong, Tat Chee Avenue, Hong Kong, China. [11] F.M. Kirby Neurobiology Center, Children's Hospital Boston, Harvard Medical School, Boston, 02115 MA, USA. [12] Department of Neurobiology, Harvard Medical School, Boston, 02115 MA, USA. [13] Department of Biological Engineering, Massachusetts Institute of Technology, Cambridge, 02139 MA, USA. [14] Division of Comparative Medicine, Massachusetts Institute of Technology, Cambridge, 02139 MA, USA. These authors contributed equally: Huihui Chen, Kin-Sang Cho, T.H. Khanh Vu. Correspondence and requests for materials should be addressed to J.C. (email: jchen@mit.edu) or to D.F.C. (email: dongfeng_chen@meei.harvard.edu)

Glaucoma affects 70 million people worldwide[1], making it the most prevalent neurodegenerative disease and a leading cause of irreversible blindness. The disease is characterized by progressive degeneration of retinal ganglion cells (RGCs) and axons. The most important risk factor for glaucoma is elevated intraocular pressure (IOP), which is thought to directly cause damage to neurons and the optic nerve. However, glaucomatous RGC and axon loss also occur in individuals with normal IOP, and patients whose IOP is effectively controlled by medical treatment often continue to suffer progressive neuron loss and visual field deterioration[2,3], suggesting mechanisms beyond pressure-mediated damage in neurodegeneration. One possibility is that pathophysiological stress, such as that induced by elevated IOP, triggers secondary immune or autoimmune responses, leading to RGC and axon damage after the initial insult is gone. To date, this remains as a hypothesis, as the molecular and cellular events underlying glaucomatous neural damage have not been identified.

Evidence suggests an autoimmune component in glaucoma[4]. Among the most direct evidence, a wide range of serum autoantibodies particularly those against heat shock proteins (HSPs) and retinal deposits of immunoglobulins, were found in glaucoma patients and animal models of glaucoma[5,6]. Moreover, inoculation of rats with human HSP27 and HSP60 induced an optic neuropathy that resembles glaucomatous neural damage[7], and elevated IOP has been reported to induce expression of HSPs in the retina, particularly RGCs[8,9]. Thus, a link among IOP elevation, HSP upregulation, and induction of anti-HSP autoimmune responses in glaucoma has been suggested; however, the roles of these events in the disease pathogenesis remain unknown.

Because the eye is an immune-privileged site, a critical question is how autoimmune responses, such as those against HSPs, are induced in glaucoma. As HSPs are among the most highly conserved proteins from bacteria to mice to humans (up to 60% identity)[10], a possibility is that the anti-HSP immune responses are induced originally by bacterial HSPs, and are reactivated by host HSPs during glaucoma. The facts that glaucoma patients exhibit increased titers of antibodies against *Helicobacter pylori* and that immunization with HSPs in rats induces glaucomatous neural damage are in line with this possibility[11]. Currently, little direct evidence is available to testify this hypothesis.

Here we show that: (1) a transient elevation of IOP is sufficient to induce CD4[+] T-cell infiltration into the retina; (2) T-cell responses are essential in the development of progressive glaucomatous neurodegeneration following IOP elevation; (3) both bacterial and human HSPs are target antigens of these T cells; and (4) HSP-specific CD4[+] T-cell responses and glaucomatous neurodegeneration are both abolished in mice raised in the absence of commensal microbial flora (germ-free (GF) mice), supporting a mechanism of bacteria sensitized T-cell responses underlying the pathogenesis of glaucoma. These observations identify a sequence of events that contribute to progressive neurodegeneration in glaucoma, and may lead to a paradigm shift for the diagnosis, prevention, and treatment of glaucoma.

## Results

**Elevation of IOP induces retinal T-cell infiltration**. To investigate if a high IOP evokes retinal immune responses, we induced IOP elevation in mice by microbead (MB) injection. As shown previously[12], a single MB injection into the anterior chamber of adult C57BL/6 (B6) mice led to a transient 3-week elevation of IOP (Fig. 1a), whereas injection of saline did not induce any significant change in IOP (Supplementary Fig. 1a). Using immunostaining of retinal flat-mounts for general T-cell marker CD3 and RGC marker Tuj1, we detected T-cell infiltration into

the ganglion cell layer (GCL) of MB-injected, but not of uninjected or saline-injected mice (Fig. 1b). Infiltrating T cells were noted at 2 weeks after MB injection (Fig. 1c), scattered throughout the retina without apparent clustering or preference to any specific quadrant. The number of T cells had declined by 4 weeks. To define the subpopulations of infiltrating T cells, we performed triple-immunolabeling with antibodies specific for CD4 or CD8 T cells, RECA1 (for blood vessels), and Tuj1. CD4[+], but not CD8[+], T cells were detected in the GCL of glaucomatous retina (Supplementary Fig. 1b). To verify T-cell retinal infiltration and define the subsets of CD4[+] T cells, we examined T-cell cytokine secretion profiles, including interferon-γ (IFN-γ) ($T_H1$), interleukin (IL)-17 ($T_H17$), IL-4 ($T_H2$), and transforming growth factor-β (TGF-β) (Treg). We detected IFN-γ secreting, but no other subsets of, CD4+ T cells in the retina. The number of infiltrating CD4[+] T cells was significantly increased in MB-injected retina compared to saline-injected controls (Fig. 1d, e). As with T-cell infiltration, significantly more CD11b[+] microglia and macrophages were detected in the retinas of mice 2–4 weeks after MB, but not after saline injection (Fig. 1f, g). These immune responses were a result of the elevated IOP because injection of fewer MB ($2.0 \times 10^6$ beads/eye), which did not elevate IOP, induced neither T-cell infiltration (Fig. 1h) nor an increase in CD11b[+] cells (Supplementary Fig. 1c) in the retina compared to saline-injected group. In contrast, correlating with IOP elevation, injection of more MB ($5.0 \times 10^6$ beads/eye) induced both T-cell infiltration (Fig. 1h) and an increase in CD11b[+] cells (Supplementary Fig. 1c). Induction of immune responses by elevated IOP was also confirmed using DBA/2J mice, which spontaneously develop increased IOP and glaucoma by 6–8 months of age[13] (Supplementary Fig. 1d, e). T-cell infiltration was detected in the retinas of 8-month-old DBA/2J mice, but not in 3-month-old mice (Fig. 1c). No B-cell infiltration was detected in the retinas in either MB-injected B6 mice or 8-month-old DBA/2J mice. Thus, elevation of IOP is followed by T-cell infiltration into the retinas.

To assess retinal neurodegeneration, we quantified RGC and axon loss in mice. As shown by RGC quantification with Tuj1 labeling, percentages of RGC and axon losses gradually increased from ~17% at 2 weeks post MB injection to ~35% by 8 weeks (Fig. 1i–k). Accordingly, RGC densities decreased significantly from ~3800 mm$^{-2}$ in saline-injected mice to ~3300 mm$^{-2}$ at 2 weeks after MB injection, and to ~2500 mm$^{-2}$ at 8 weeks after MB injection. This result was further verified by immunolabeling for Brn3a (Supplementary Fig. 1f,g). Notably, although elevation of IOP lasted only transiently for 3 weeks in the MB-injected mice, RGC and axon loss continued to 8 weeks after MB injection, the longest time point that the mice were monitored. These results show that a transient elevation of IOP induces T-cell infiltration into the retina and a prolonged period of retinal neurodegeneration, even after the IOP has returned to the normal level.

**T cells mediate prolonged retinal neurodegeneration**. To determine the role of T cells in elevated IOP-induced retinal neurodegeneration, we used mice deficient in both T and B cells (*Rag1*$^{-/-}$ mice), only T cells (*TCRβ*$^{-/-}$ mice), or only B cells (*Igh6*$^{-/-}$ mice)[14–16]. Injection of MB into these immune-deficient mice resulted in a transient elevation of IOP that had the same magnitude and kinetics as that observed in B6 mice (Supplementary Fig. 2a) and a similar loss of RGCs and axons at 2 weeks as seen in B6 mice (Fig. 2a, b). However, glaucomatous *Rag1*$^{-/-}$ mice did not exhibit continued loss of RGCs or axons from 2 to 8 weeks post MB injection when IOP had returned to the normal range, whereas B6 and *Igh6*$^{-/-}$ mice did. In *TCRβ*$^{-/-}$ mice, which lack CD4[+] αβT cells but are compensated by expansion of

γδ and natural killer T cells and B cells, no significant further loss of RGCs was detected between 2 and 8 weeks after MB injection; whereas, an attenuated albeit significant loss of axons was observed at 8 weeks. The data suggest a primary role for CD4+ αβT cells, with a possible involvement of other immune cells, in late glaucomatous axon damage. These findings support a two-phase neurodegeneration in glaucoma: the initial phase when IOP is elevated and the prolonged (progressive) phase after IOP has returned to the normal range. Neither T- nor B-cell deficiency attenuates the initial phase of neural damage, but T cells are an essential player in the prolonged phase of glaucomatous RGC and axon degeneration.

To further define the subsets of systemic CD4+ T cells that were activated under elevated IOP, we examined cytokine secretion profiles of CD4+ T cells in the spleens and eye draining (cervical) lymph nodes (LNs) by flow cytometry. Unlike that seen in the retina, the frequencies of CD4+ T cells that expressed IFN-γ (T$_H$1), IL-17 (T$_H$17), IL-4 (T$_H$2), or TGF-β (Treg) were all significantly increased in both the spleen and the cervical LN of mice at 2 weeks post MB injection compared to saline-injected mice (Supplementary Fig. 2b, c). Thus, elevated IOP induced enhanced systemic reactivity of T cells, without any bias toward a specific subset of CD4+ T cells.

We investigated if CD4+ T cells play a causal role in progressive neurodegeneration in glaucoma by adoptively

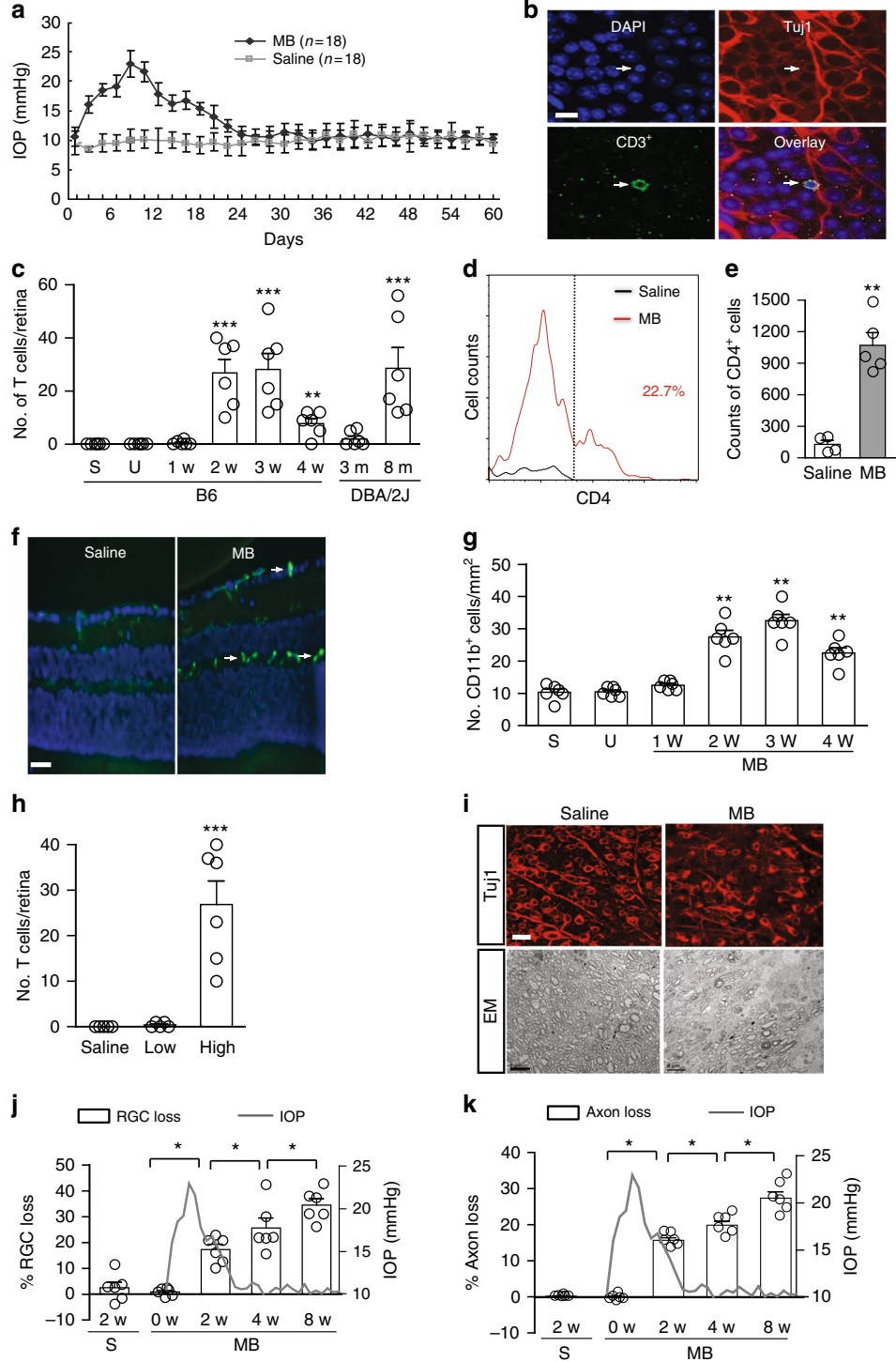

transferring T cells from glaucomatous B6 mice into $Rag1^{-/-}$ mice. As all subtypes of CD4$^+$ T cells were activated following IOP elevation, total CD4$^+$ T-cell populations from the spleens of B6 mice, which had received MB (glaucomatous) injection 14 days earlier, were transferred; CD4$^+$ T cells isolated from naive or saline-injected mice served as controls. Cells were injected via the tail vein into $Rag1^{-/-}$ mice that had received MB injection 14 days earlier (on the same day when the donor mice had received MB injections) (Fig. 2c). Two weeks after CD4$^+$ T-cell transfer (or 4 weeks after MB injection), MB-injected $Rag1^{-/-}$ recipients, which had not received any CD4$^+$ T-cell transfer (U) or that received CD4$^+$ T cells from saline-injected (S) normal IOP B6 mice showed no retinal T-cell infiltration (Fig. 2d, f). As expected, these $Rag1^{-/-}$ mice exhibited ~15% RGC and axon loss as a result of the elevated IOP-induced initial phase of neural damage (Fig. 2g, h). In contrast, MB-injected $Rag1^{-/-}$ recipients that had received a CD4$^+$ T-cell transfer from glaucomatous (MB-injected) B6 mice showed retinal T-cell infiltration (Fig. 2f) and a significant further increase in RGC and axon loss compared to the uninjected (U) and saline-injected (S) control mice (Fig. 2d–h). Induction of IOP elevation in recipient mice was required for T-cell transfer-induced neurodegeneration because injection of T cells from glaucomatous mice into $Rag1^{-/-}$ mice with normal IOP did not cause further RGC and axon degeneration after week 2 (Supplementary Fig. 2d, e). Injection of total IgG from the MB-injected B6 mice induced no significant increase in RGC and axon loss in MB-injected $Rag1^{-/-}$ mice (Supplementary Fig. 2f, g). These results indicate that conditioned CD4$^+$ T cells are sufficient to drive the prolonged phase of retinal neurodegeneration in glaucomatous mice.

**IOP elevation activates T-cell responses to HSPs**. To define the autoantigens that stimulate T-cell activation in glaucoma, we searched for proteins that are expressed or upregulated in RGCs following IOP elevation. Since autoantibodies to HSPs, particularly HSP27 and HSP60, had previously been detected in patients and animal models of glaucoma[8,17], we speculated that these HSPs are involved as pathogenic autoantigens, which may be expressed at a low level under the normal condition but are upregulated in RGCs under glaucoma. The levels of HSP27 and HSP60 proteins were low in the retinas of B6 mice with normal IOP, but were upregulated three- to fourfold 1–8 weeks after MB injection (Fig. 3a; Supplementary Fig. 3a–c). HSP27 upregulation was found primarily in the GCL (Supplementary Fig. 2h), both associated with RGC membranes and outside of RGC bodies (Fig. 3b). This is consistent with the report that HSP27 is also upregulated in astrocytes following elevated IOP and be released

extracellularly under stress[18,19]. HSP60 signal was seen to be scattered between the GCL and INL in MB-injected retina (Supplementary Fig. 3d). Significantly higher levels of anti-HSPs, particularly anti-HSP27 autoantibody, were detected in the sera of MB-injected B6 mice than in the serum of normal IOP mice (Fig. 3c).

To test if HSP27 acts as an antigen participating in elevated IOP-induced T-cell responses, we assessed delayed-type hypersensitivity (DTH) responses in mice. Significant induction of a HSP27-specific DTH was observed 2 and 8 weeks after MB injection in B6 mice when compared to saline-injected mice (Fig. 3d, e). More CD4$^+$ T-cell infiltration was detected in the ear sections of MB-injected than in saline-injected B6 mice (Supplementary Fig. 4a). In agreement with the absence of elevated IOP-induced T-cell responses in $Rag1^{-/-}$ and $TCR\beta^{-/-}$ mice, HSP27-specific DTH was not evoked in these mice (Fig. 3d). The DTH response was specific to HSP27 because challenge with control antigens, such as human myelin basic protein (MBP) or interphotoreceptor retinoid-binding protein (IRBP), did not induce a DTH in MB-injected B6 mice. Induction of HSP27-specific T-cell responses by elevated IOP was also verified using an enzyme-linked immunospot (ELISPOT) assay: HSP27 stimulation of splenocytes from mice at 2 and 8 weeks after MB injection induced a ~10 times higher frequency of IFN-γ-secreting T cells than the splenocytes of saline-injected B6 mice; in contrast, the frequencies of IFN-γ-secreting T cells in MB-injected $Rag1^{-/-}$ and $TCR\beta^{-/-}$ mice were comparable with those of saline-injected B6 mice (Fig. 3f). Stimulation with control antigen MBP did not increase the frequency of IFN-γ-secreting T cells, further corroborating an induction of HSP27-specific T-cell responses. These T-cell responses were induced by elevated IOP, but not by the presence of MBs, as injection of fewer MB, which did not cause IOP elevation did not lead to an increased frequency in IFN-γ-secreting cells (Supplementary Fig. 4b). Moreover, a similar increase in the frequencies of IFN-γ-secreting T cells was observed in splenocytes taken from 10-month-old DBA/2J mice compared to those from 3-month-old DBA/2J mice or age-matched (10-month-old) control B6 mice (Supplementary Fig. 4c). Thus, elevated IOP-induced T-cell responses are specific to HSPs.

**HSP-specific T cells augment glaucomatous neurodegeneration**. To examine if HSP27-specific IFN-γ-secreting T cells infiltrate the retina and contribute to glaucomatous pathogenesis, we performed fluorescence-activated cell sorting analysis for HSP27-stimulated retinal cell cultures following a gating strategy shown in Supplementary Fig. 5. At 2 weeks post MB injection,

**Fig. 1** A transient IOP elevation induces T-cell infiltration and progressive RGC and axon loss. **a** IOP levels in MB- or saline-injected B6 mice ($n = 18$/group). **b** Retinal T-cell infiltration post MB injection. Retinal flat-mounts from mice 3 weeks after MB injection were stained with Tuj1 (red), an anti-CD3 antibody (green; arrow), and DAPI (blue). Scale bar: 10 μm. **c** Quantification of CD3$^+$ T cells in retinas of uninjected mice (U), mice 3 weeks after saline injection (S), or 1–4 weeks (w) after MB injection ($n \geq 6$/group), and DBA/2J mice at 3 or 8 ($n = 6$/group) months (m) old. **$P < 0.01$, ***$P < 0.001$ as compared to saline-injected group or between 3- and 8-month-old DBA/2J mice. **d** Flow cytometry plot of retinal cells double-immunolabeled for IFN-γ and CD4; cell count was gated for IFN-γ-secreting cells. **e** Quantification of CD4$^+$ T cells by flow cytometry 2 weeks post saline or MB injection ($n = 5$/group). Mice were perfused with saline. CD4$^+$ cell counts were obtained by multiplying total cells recovered from the retina with percentage of CD4$^+$ cells. **$P < 0.01$ by $t$-test ($n = 6$/group). **f** Representative photomicrographs of retinal sections labeled with anti-CD11b (green; arrow) and counter-stained with DAPI (blue). Scale bar: 20 μm. **g** Quantification of CD11b$^+$ cells at 1–4 weeks post MB injection, 3 weeks post saline (S) injection, and in uninjected (U) mice. **$P < 0.01$ as compared to saline-injected group ($n = 6$/group). **h** Quantification of infiltrated T cells in the retinas of mice 2 weeks after a saline, low (Low; $2.0 \times 10^6$ beads/eye) or high (High; $5.0 \times 10^6$ beads/eye) dose of MB injection. ***$P < 0.001$ as compared to saline-injected group ($n = 6$/group). **i** Representative Tuj1-stained epifluorescence photomicrographs of retinal flat-mounts (Tuj1) and electron microscopy (EM) of optic nerve cross sections from mice 8 weeks after saline or MB injection. Scale bar for Tuj1 and EM: 10 and 50 μm, respectively. **j, k** Progressive loss of RGCs and axons in MB-injected mice. RGCs were counted in Tuj1-stained retinal flat-mounts and axons in optic nerve cross sections. Shown are percentage of RGC (**h**) and axon (**i**) loss. The latter is overlaid with changes in IOP levels in MB-injected eyes. *$P < 0.05$, **$P < 0.01$, ***$P < 0.001$ by ANOVA as compared to saline-injected mice ($n = 6$/group)

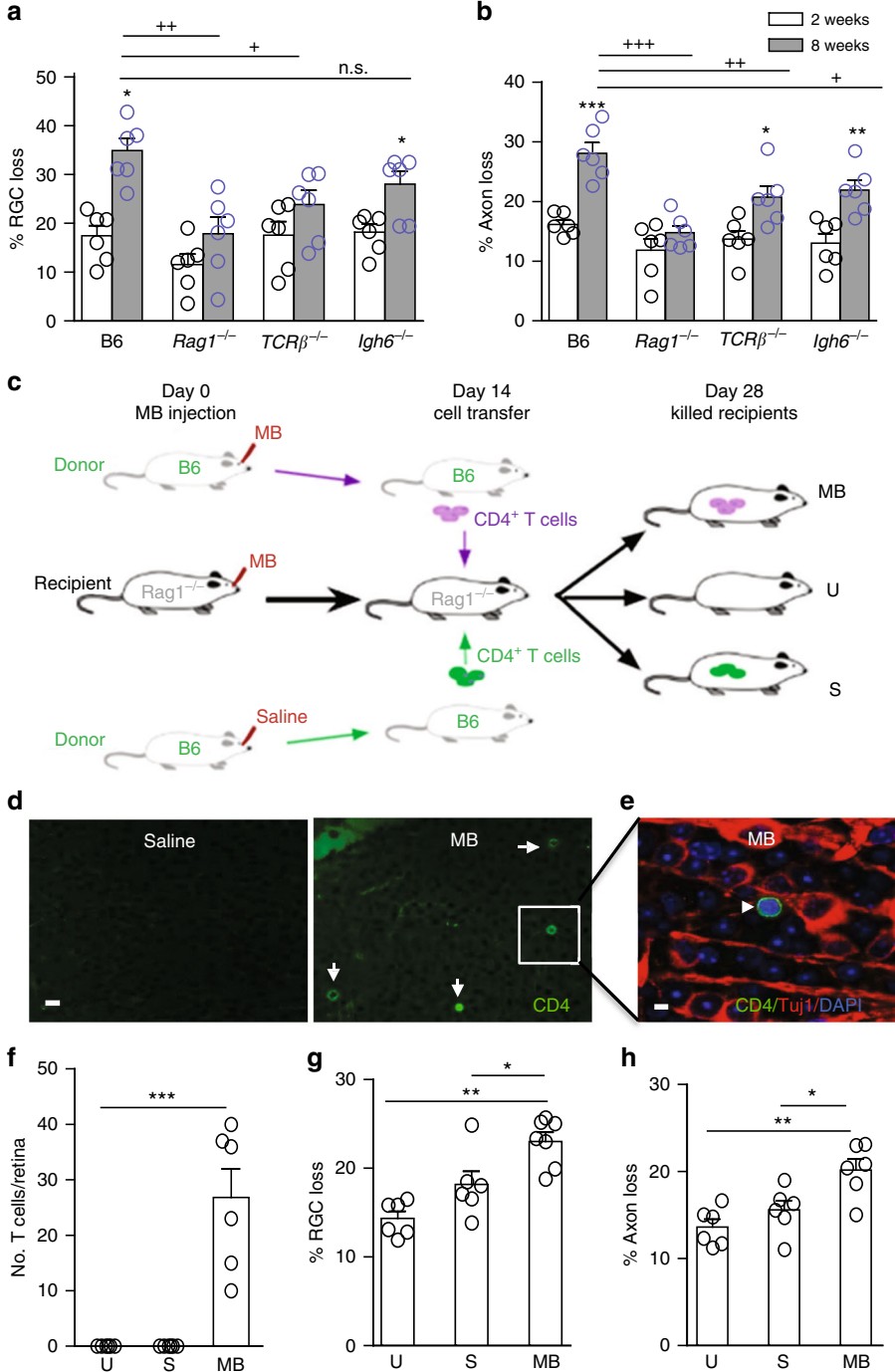

**Fig. 2** T cells are required for the prolonged retinal neurodegeneration. **a**, **b** Quantification of RGC (**a**) and axon (**b**) loss in B6, *Rag1*$^{-/-}$, *TCRβ*$^{-/-}$, and *Igh6*$^{-/-}$ mice at 2 (white box) and 8 (gray box) weeks after anterior chamber MB injection. n.s. nonsignificant, $^+P < 0.05$, $^{++}P < 0.01$, $^{+++}P < 0.001$ by ANOVA comparing between B6 and indicated mutant mice at 8 weeks post MB injection; $^*P < 0.05$, $^{**}P < 0.01$, $^{***}P < 0.001$ by ANOVA comparing between 2 and 8 weeks post MB injection of mice with the same genotype ($n = 8$/group). **c** Scheme of adoptive CD4$^+$ T-cell transfer. Both donor B6 and recipient *Rag1*$^{-/-}$ (Rag1$^{-/-}$) mice were injected with MB or saline into the anterior chamber. Fourteen days later, CD4$^+$ T cells were isolated from the spleens of donor mice and injected via tail vein into Rag1$^{-/-}$ mice that had received MB injection 14 days earlier. **d**–**h** Fourteen days post cell transfer, T-cell infiltration and RGC and axon loss in *Rag1*$^{-/-}$ mice were analyzed. Retinal flat-mounts were stained with an anti-CD4 antibody (green), Tuj1 (red), and DAPI (blue). Shown are anti-CD4-stained (**d**) and overlay (**e**, from inset of **d**) images of retinal flat-mounts taken from *Rag1*$^{-/-}$ recipient mice receiving CD4$^+$ T cells from saline- or MB-injected B6 mice. Arrows point to CD4$^+$ cells. Scale bar: 25 μm (**d**) and 10 μm (**e**). Quantification of infiltrated T cells (**f**), and RGC (**g**) and axon (**h**) loss in glaucomatous *Rag1*$^{-/-}$ recipient mice that received no CD4$^+$ T cells (U uninjected) or donor CD4$^+$ T cells from saline-injected (S or normal IOP donor) and MB-injected (MB or glaucomatous donor) B6 mice 2 weeks post cell transfer. $^*P < 0.05$; $^{**}P < 0.01$ by ANOVA ($n = 6$/group)

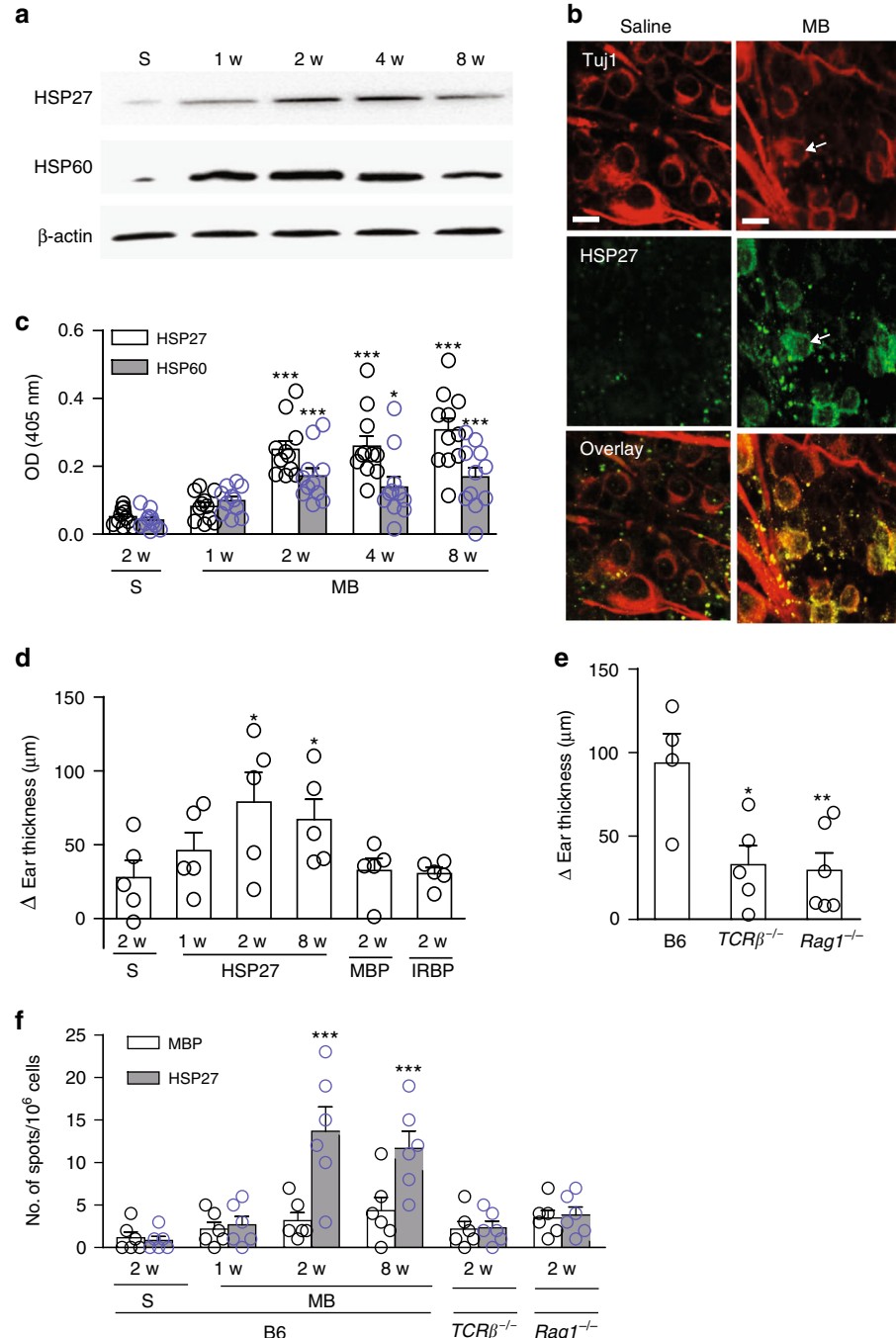

**Fig. 3** Elevated IOP stimulates T-cell responses to HSPs. **a** Induction of HSP27 and HSP60 expression in the retinas following MB injection. B6 mice were injected with MB or saline and 1, 2, 4, and 8 weeks later retinas were harvested and homogenized. Cell homogenates were fractionated on SDS-PAGE and blotted with anti-HSP27, anti-HSP60, and anti-β-actin. Shown are representative western blots. **b** Upregulation of HSP27 in the glaucomatous retina. Retinal flat-mounts were stained with Tuj1 (red) and anti-HSP27 (green). Shown are representative epifluorescence photomicrographs taken from mice 4 weeks post injection of saline or MB. Note the extracellular and membrane-associated HSP27 signals in the MB-injected retina. Scale bar: 10 μm. **c** ELISA quantification of serum levels of antibodies specific for HSP27 or HSP60 in B6 mice 1, 2, 3, and 4 weeks after MB injection or 2 weeks after saline (S) injection. *$P < 0.05$, ***$P < 0.001$ by ANOVA comparing to saline-injected group ($n = 10$/group). **d**, **e** DTH assays for T-cell responses to HSP27. Shown are ear thickness among B6 mice under indicated conditions (**d**) or among B6, $Rag1^{-/-}$, and $TCR\beta^{-/-}$ mice 2 weeks after MB injection were compared (**e**). B6, $Rag1^{-/-}$, and $TCR\beta^{-/-}$ mice were injected with saline or MB in the anterior chamber of the eye. One, two, and eight weeks later mice were challenged with saline, HSP27, MBP, or IRBP, and ear thickness was measured 24 h later. *$P < 0.05$ by ANOVA compared to saline (**d**) or to HSP27 challenged B6 mice (**e**) ($n \geq 4$/group). **f** Frequencies of HSP27-specific T-cell responses. B6, $Rag1^{-/-}$, and $TCR\beta^{-/-}$ mice were injected with saline or MB in the anterior chamber of the eye. One, two, and eight weeks later splenocytes were stimulated with HSP27 or MBP and the frequencies of IFN-γ-secreting cells were measured by ELISPOT. ***$P < 0.001$ by ANOVA as compared to saline- and MB-injected mice ($n \geq 6$/group)

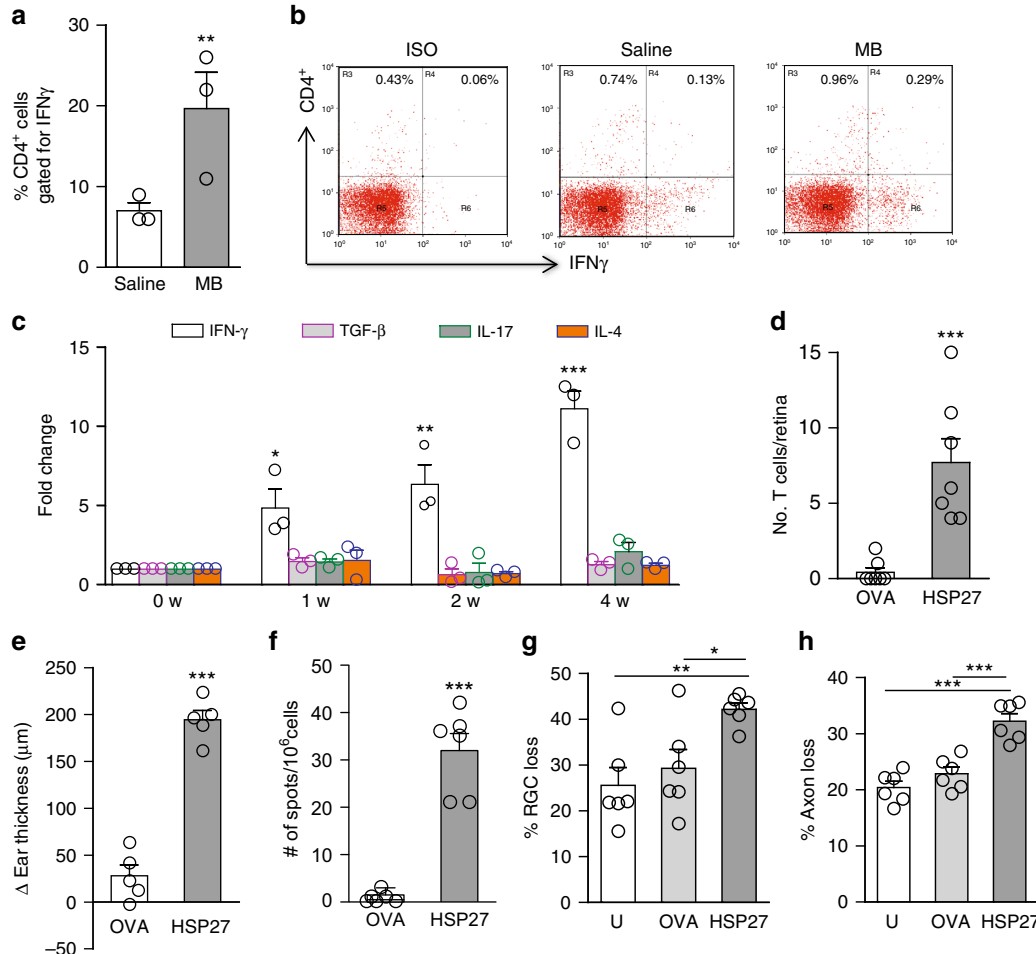

**Fig. 4** HSP-specific T cells infiltrate the retinas and augment glaucomatous neurodegeneration. **a**, **b** Infiltration of HSP27-specific T cells into the glaucomatous retina. Retinal cells from mice at 2 weeks post saline or MB injection were stimulated by HSP27 in culture, stained for CD4 and IFN-γ, and assayed by flow cytometry. Shown are frequencies of INF-γ+ -cells in CD4+ gated cells (**a**) and representative isotype control (Iso) and anti-CD4 vs. anti-IFN-γ staining profiles (**b**) from the retina of saline- and MB-injected mice. The numbers indicate percentages of cells in the gated regions. Note the increased number of INF-γ+ cells among CD4+ cells in MB-injected retina compared to saline-injected retina. **$P < 0.01$ ($n = 3$/group; each group was pooled from 5 mice). **c** qPCR quantification of IFN-γ, TGF-β, IL-17, and IL-4 transcripts in the mouse retinas before (0) and at 1, 2, and 4 weeks post MB or saline injection. **$P < 0.01$, ***$P < 0.001$ by ANOVA ($n = 3$/group). **d** Quantification of CD4+ T-cell numbers in retinal flat-mounts of glaucomatous $Rag1^{-/-}$ mice that received an injection of GFP+/CD4+ T cells from HSP27- or OVA-immunized mice. ***$P < 0.001$ ($n = 6$/group). **e**, **f** T-cell responses in ovalbumin (OVA)- or HSP27-immunized mice. B6 mice were immunized with HSP27 or OVA, and 2 weeks later, mice were challenged with HSP27 by intradermal injection in the ear followed by measurement of ear thickness 24 h later (**e**) or mouse splenocytes were isolated, stimulated with HSP27 and then IFN-γ-secreting cells were detected by ELISPOT (**f**). ***$P < 0.001$ ($n = 6$/group). **g**, **h** Greater loss of RGCs (**g**) and axons (**h**) in WT recipient mice received adoptive T-cell transfer from HSP27-immunized mice. B6 mice were immunized with HSP27 or ovalbumin (OVA) and 2 weeks later CD4+ T cells were isolated from spleens of immunized mice and unimmunized (U) mice, and adoptively transferred into B6 mice that had been injected with MB 2 weeks earlier. Two weeks after cell transfer, retinal flat-mounts of recipient mice were stained with Tuj1, and RGCs were counted. **$P < 0.01$, ***$P < 0.001$ by ANOVA ($n = 6$/group)

~1000 CD4+ T cells were noted in a glaucomatous mouse retina (Fig. 1e). Supporting retinal infiltration of HSP27-specific IFN-γ-secreting T cells, HSP27 stimulation induced an approximately threefold higher frequency of IFN-γ+CD4+ T cells from the retinal cultures of B6 mice than from saline-injected mice (Fig. 4a, b; Supplementary Fig. 5); whereas, the other three subtypes of CD4+ T cells ($T_H2$, $T_H17$, and Treg) were not detectable. Results of quantitative PCR also detected the induction of IFN-γ expression, but a lack of TGF-β, IL-17, and IL-4 transcripts, in the glaucomatous retinas at 1, 2, and 4 weeks after MB injection (Fig. 4c). To corroborate the above finding, GFP transgenic mice were immunized with HSP27 or ovalbumin (OVA), and CD4+ T cells from immunized mice were adoptively transferred into glaucomatous $Rag1^{-/-}$ mice (which had received an MB injection

2 weeks earlier). Approximately 15-fold more GFP+ T cells were detected in the retinas of recipient $Rag1^{-/-}$ mice after T-cell transfer from HSP27- than OVA-immunized donors (Fig. 4d), further demonstrating HSP27-specific T-cell infiltration into the glaucomatous retinas.

We then tested if HSP27-specific T cells can induce progressive glaucomatous neural damage. B6 mice were immunized with HSP27 or OVA. Successful immunization was confirmed by DTH and ELISPOT assays (Fig. 4e, f). CD4+ T cells were then isolated from spleens of the immunized or unimmunized mice and adoptively transferred into glaucomatous B6 mice (injected with MB 2 weeks earlier). Recipient B6 mice that received CD4+ T cells from the HSP27-immunized mice exhibited a significantly greater loss of RGCs and axons compared to mice that received

T cells from unimmunized or OVA-immunized mice (Fig. 4g). Thus, HSP27-specific CD4$^+$ T cells mediate the prolonged phase of RGC and axon degeneration.

**Absence of glaucomatous neural damage in GF mice**. A key remaining question here is how T-cell responses to HSPs are induced by elevated IOP, especially given that the retina is an immune-privileged site. HSPs are highly conserved from bacteria to mice to humans[20,21]; mice and humans are exposed to commensal microflora after birth and occasionally are infected by pathogenic bacteria. We hypothesized that mice harbor memory T cells to bacterial HSPs that can be activated by host HSPs through molecular mimicry when the blood-retinal barrier is compromised by elevated IOP. If this is the case, we expected that T cells from glaucomatous mice would react to both bacterial and human HSPs. Indeed, following stimulation with either human HSP27 and HSP60 or *Escherichia coli* HSP24 and HSP60, we observed similar increases in the frequencies of IFN-γ-secreting cells in splenocyte cultures of MB-injected mice as compared to saline-injected mice (Fig. 5a).

We reasoned that mice raised in the absence of microbial colonization would exhibit attenuated or absent HSP-specific T-cell responses and retinal neurodegeneration following IOP elevation. We therefore examined T-cell responses and glaucomatous neurodegeneration in GF Swiss Webster mice, which were readily available from a commercial source. Anterior chamber injection of MB in GF Swiss Webster mice induced IOP elevation with similar kinetics and magnitude to conventionally colonized (specific pathogen-free [SPF]) Swiss Webster mice (Supplementary Fig. 6a). HSP27 upregulation in the GCL of both SPF and GF mice following IOP elevation was verified by immunolabeling, but no apparent differences between the SPF and GF groups, before or after IOP elevation, were noted (Supplementary Fig. 6b). Elevated IOP-induced retinal T-cell infiltration (Fig. 5b) and HSP27- and HSP60-specific T-cell responses (Fig. 5c, d) were detected in MB-injected SPF but not in GF mice. Correspondingly, glaucomatous RGC and axon damage was observed in SPF but not in GF mice, when examined either 4 or 8 weeks following MB injection (Fig. 5e, f). We then examined "altered Schaedler flora" (ASF) Swiss Webster mice, which were colonized with eight defined bacterial species[22]. We noted significant RGC and axon loss following MB injection in ASF mice, although the magnitude of loss was significantly attenuated compared to MB-injected SPF mice (Fig. 5e, f). These results suggest that induction of a full spectrum of HSP-specific T-cell responses and retinal neurodegeneration requires pre-exposure to diverse microbial flora, rather than a specific bacterial species.

To verify the above finding using the inherited mouse model of glaucoma, we derived/created GF DBA/2J mice. DBA/2J mice housed under SPF or GF conditions both naturally developed high IOP with a peak of ~22 mm Hg by 8–10 months of age (Supplementary Fig. 6c), consistent with previous reports[23]. The SPF DBA/2J mice underwent the expected progressive RGC and axon loss: ~25% by 8–10 months and ~50% by 12 months of age, whereas no neural loss was detected in GF DBA/2J mice as late as 12 months of age (Supplementary Fig. 6d; Fig. 5g, h). In agreement with the above finding, both RGC and axon counts remained stable from 3 to 12 months of age, showing the absence of neurodegeneration in GF DBA/2J mice. Collectively, these results indicate a need for prior exposure to commensal microbial flora in the induction of both HSP-specific T-cell responses as well as RGC and axon loss following IOP elevation.

**HSP-specific T cells are increased in patients with glaucoma**. To investigate the possible involvement of HSP-specific T-cell responses in human glaucoma, we compared the frequencies of HSP27- and HSP60-specific T cells and the levels of auto-antibodies between normal subjects and patients with primary open-angle glaucoma (POAG) or normal tension glaucoma (NTG). The frequencies of HSP27- and HSP60-responsive T cells were over fivefold higher in both POAG and NTG patients than in age-matched healthy individuals (Fig. 6a, b). Patients with retinal detachment or traumatic skin injuries did not have any significant increase in HSP27- or HSP60-reactive T cells. In addition, the titers of HSP27- and HSP60-specific IgGs were approximately twofold higher in both POAG and NTG patients than in healthy individuals (Fig. 6c, d), which was consistent with a previous report[24]. Thus, the levels of HSP27- and HSP60-reactive T cells and antibodies are also elevated in patients with POAG and NTG.

## Discussion

This is the first report that, to our knowledge, describes an unexpected link and the sequential roles of elevated IOP, intact commensal microflora, and activation of T-cell responses in the pathogenesis of glaucoma. Our comprehensive studies employed both inducible and genetic mouse models of glaucoma, immunodeficient mice, GF mice, clinical samples, and adoptive T-cell transfer, and assessed T-cell infiltration and retinal neurodegeneration. We showed that (1) a transient elevation of IOP by MB injection is sufficient to induce T-cell infiltration into the retina and a prolonged phase of retinal neurodegeneration, (2) T cells are required for mediating the prolonged retinal neurodegeneration and are specific for HSPs, and (3) induction of HSP-specific T cells and glaucomatous retinal neurodegeneration by elevated IOP require exposure to commensal microflora. Together, these findings are compelling as they suggest the essential involvement of a T-cell-mediated mechanism in the pathogenesis of glaucoma. A similar mechanism likely operates during the development of glaucoma in humans as HSP-specific T cells are more than fivefold higher in POAG and NTG patients than age-matched healthy individuals.

Our study is the first demonstration of direct induction of HSP-specific T-cell responses by an elevation of IOP. HSP-specific T-cell responses have been proposed to contribute to NTG as HSP immunization elicits glaucomatous RGC loss in rats[7]. Here we show that elevated IOP induces two phases of retinal damage: the acute phase correlates with the IOP elevation, likely mediated by physical stress; the prolonged retinal degeneration, which continues even after IOP has returned to normal, is mediated by T cells. Supporting this notion, mice deficient in T cells, but not B cells, displayed a dramatically attenuated RGC and axon damage. Adoptive transfer of CD4$^+$ T cells, but not total IgG antibodies, from diseased mice restored the progressive retinal neurodegeneration in *Rag1*$^{-/-}$ recipients. Adoptive transfer of CD4$^+$ T cells from HSP27- but not OVA-immunized mice augmented the progressive retinal neurodegeneration in MB-injected B6 recipients. Similarly, adoptive transfer of CD4$^+$ T cells from HSP27-immunized mice into MB-injected *Rag1*$^{-/-}$ recipient mice induced 16 ± 2.1% (*n* = 5/group) more RGC loss 2 weeks after cell transfer as compared to those receiving T cells from OVA-immunized mice. The kinetics of T-cell infiltration into the retinas and their specificity for HSPs lends further support for their involvement in the progressive retinal degeneration. Furthermore, accumulating evidence indicates an essential role for commensal microflora in T-cell activation. Lack of HSP-specific T-cell responses and neural damage in GF mice offers the first and compelling evidence supporting that mere elevation of IOP does not directly attribute to progressive neurodegeneration. It is the subsequent event, involving T-cell responses, which have

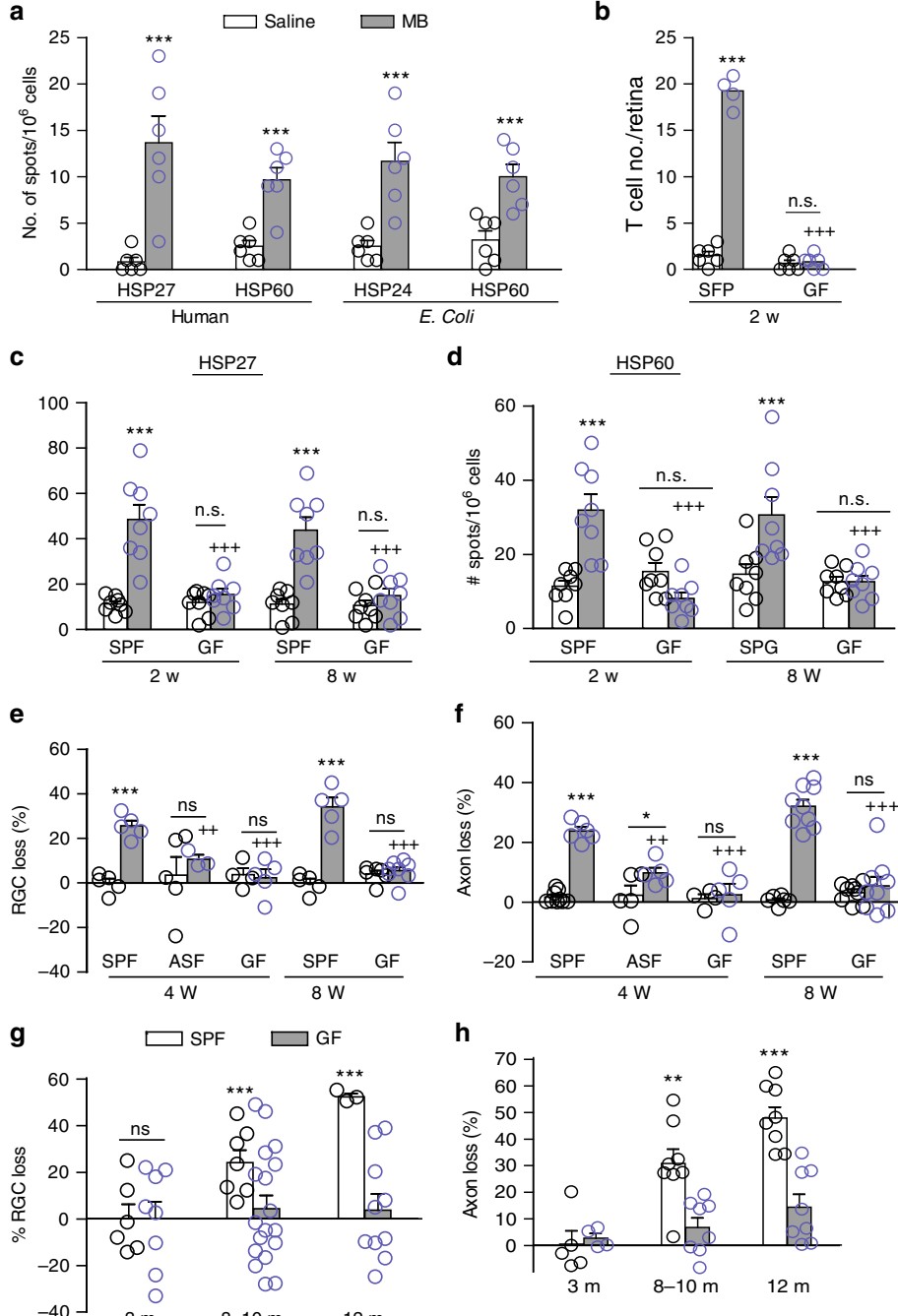

**Fig. 5** HSP-specific T-cell responses and retinal damage are absent in germ-free mice. **a** Frequencies of T-cell responses to human and bacterial HSPs. Splenocytes from mice at 2 weeks post saline or MB injection were stimulated with human HSP27 and HSP60 or *E. coli* HSP24 and HSP60, and the frequencies of IFN-γ-secreting cells were assayed by ELISPOT (*n* = 6/group). **b** Quantification of infiltrated T cells in the retinas of SPF and GF SW mice at 2 weeks after saline or MB injection (*n* = 6/group). **c**, **d** Frequencies of HSP27 (**c**)- and HSP60 (**d**)-specific T cells in GF and SPF Swiss Webster mice (*n* ≥ 8/group). Splenocytes from GF and SPF Swiss Webster mice at 2 and 8 weeks post MB or saline injection were stimulated with human HSP27 or HSP60, and the frequencies of IFN-γ-secreting cells were assayed by ELISPOT. n.s. not significant. **e**, **f** Comparison of RGC (**e**) and axon (**f**) loss at 4 and 8 weeks after MB injection in SPF, ASF, and GF Swiss Webster mice. *$P$ <0.05; ***$P$ < 0.001 over saline-injected group; ++$P$ < 0.01; +++$P$ < 0.001 over MB-injected SPF mice at the indicated time points. n.s. not significant (*n* = 8/group). **g**, **h** Comparison of RGC (**g**) and axon (**h**) loss in SPF and GF DBA/2J mice at 3 (*n* = 16), 8–10 (*n* = 24), and 12 (*n* = 12) months of age. n.s. not significant; **$P$ < 0.01, ***$P$ < 0.001 by ANOVA

been pre-sensitized by commensal microflora, that mediates progressive glaucomatous neurodegeneration. Finally, we show that HSP-specific T-cell responses occur not only in glaucomatous mice but also in patients with glaucoma. The induction of HSP-specific T cells is not a result of general stress responses as patients with retinal detachment or traumatic skin diseases did not induce HSP-specific T-cell responses. These findings show

that IOP elevation triggers secondary anti-HSP CD4[+] T-cell responses that mediate a prolonged retinal neurodegeneration, providing an explanation for continued neurodegeneration in patients with normal or perfectly controlled IOP.

Previous studies have speculated a connection between glaucoma and bacterial infections, such as by *Helicobacter pylori*[25]. The notion that a process of microbiota-dependent activation of

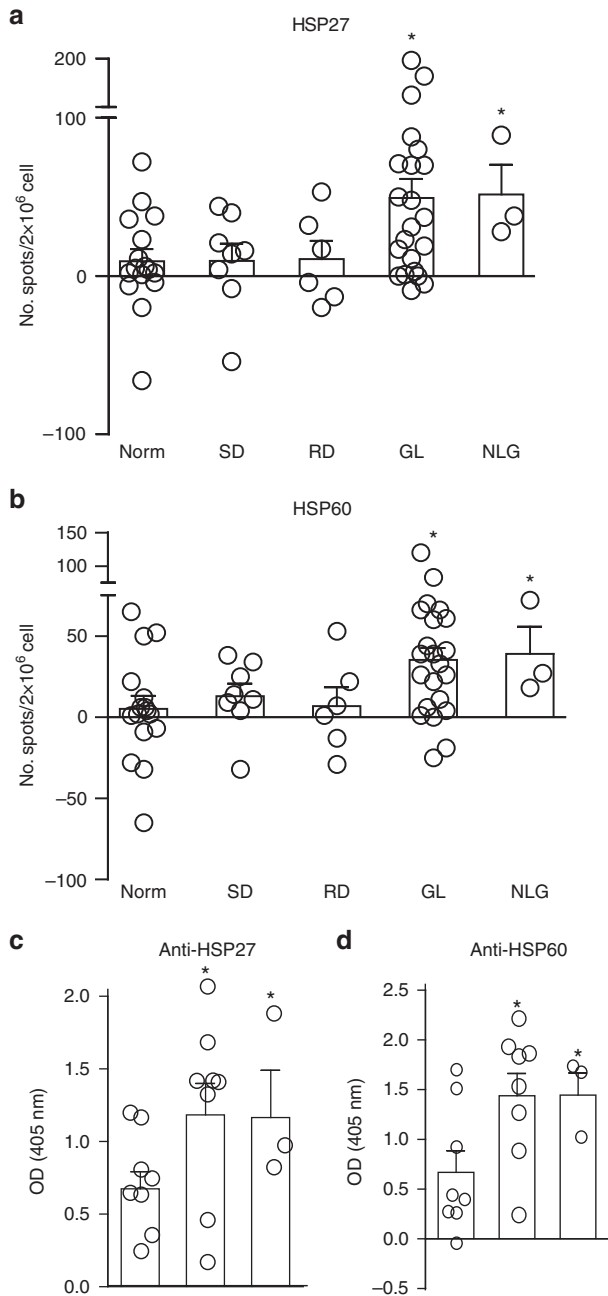

**Fig. 6** T cells are involved in retinal neurodegeneration in human glaucoma patients. **a**, **b** Comparison of frequencies of HSP27- and HSP60-specific T cells among healthy individuals and patients with POAG or other diseases. Peripheral blood mononuclear cells from patients with POAG ($n = 18$), NTG ($n = 3$), retinal detachment (RD; $n = 6$), skin injuries (SD; $n = 8$), and age-matched healthy controls (Norm; $n = 16$) were stimulated with HSP27 (**a**) or HSP60 (**b**). Two days later, the frequencies of IFN-γ-secreting cells were determined by ELISPOT. *$P < 0.05$ by ANOVA. **c**, **d** Comparison of serum levels of HSP27- and HSP60-specific IgGs between healthy individuals and patients with POAG or NTG. Sera from POAG ($n = 18$), NTG ($n = 3$), and age-matched healthy individuals (Norm, $n = 16$) were assayed for HSP27- and HSP60-specific IgG by ELISA. *$P < 0.05$ by ANOVA

T cells may precede the clinical onset of a disease in the eye is supported by a recent study in a uveitis animal model[26]. Using transgenic mice expressing a T-cell receptor (TCR) specific to IRBP in the retina, the authors showed that commensal microbiota is required for the activation of transgenic T cells to cause

uveitis. Although both our and the previous studies demonstrated the requirement of commensal microflora and autoreactive T cells in inflammatory diseases in the eye, the two studies differ in several aspects. The previous study examined how autoreactive transgenic T cells were activated to cause uveitis—a model of experimental encephalomyelitis; whereas, our study examined how endogenous T cells could be activated to perpetuate glaucomatous retinal neurodegeneration. Furthermore, our study identified both bacterial and host HSPs as likely key natural antigens, and showed that commensal microflora induces HSP-specific memory T cells, which are then activated by host HSPs induced in the retina after IOP elevation. Our study also showed that elevated IOP-induced degeneration of RGCs and axons was partially attenuated in ASF mice with eight defined bacteria, suggesting that induction of CD4$^+$ T-cell responses by HSPs requires a diverse flora rather than a specific bacterium. Thus, our study shows how physical stress, HSPs, microbial flora, and T cells interact in the pathogenesis of glaucomatous neurodegeneration.

Induction of HSP expression has been documented to associate with various pathological conditions in the retina and brain[27,28], and similar mechanisms may underly other causes of neurodegeneration. Intracellular expression of HSPs is thought to protect neurons and axons from stress-induced damage[29,30] and promotes nerve regeneration[31]. However, it has been reported that membrane-bound and extracellular HSPs, which are likely to be "seen" by antigen-presenting cells and T cells, elicit immune responses of the adaptive or innate immune system[32]. Our results show that elevation of IOP upregulates membrane-bound and extracellular HSPs in the GCL, subsequently leading to immune-mediated neural damage through activating HSP-specific CD4$^+$ T cells, which are originally induced by microbial HSPs. As microbes and humans share significant homologies in many other proteins, it is likely that HSPs are not the only cross-reactive antigens involved. The same mechanism of molecular mimicry may activate T cells, which are originally induced by other microbial proteins. Consequently, this may lead to induction of autoimmune responses under stress conditions. T-cell sensitization and progressive neurodegeneration were also noted in other conditions involving HSP upregulation, including retinal ischemia and optic nerve trauma (unpublished data). Identification of CD4$^+$ T-cell responses in glaucoma opens the possibility of targeting T cells in the retina as a treatment to halt the progressive RGC and axon degeneration and vision loss.

Induction of retinal damage by adoptive transfer of CD4$^+$ T cells is associated with local injury/inflammation, such as that induced by elevated IOP. We noted that neurodegeneration was observed following T-cell transfer only into high IOP mice, but not those with a normal IOP, suggesting that local inflammation is required for T-cell infiltration into the retina. This offers an explanation for why glaucomatous neural damage is limited to the retina and optic nerve as elevated IOP selectively causes inflammation in these areas. This notion is supported by the observation that peripheral administration of lipopolysaccharide exacerbated neuron loss in glaucoma[33]. Another report, however, showed that neurodegeneration could be induced in mice even with a normal IOP 8 weeks or longer after adoptive transfer of T cells isolated from genetic mouse models of glaucoma[34]. It suggests that activated T cells of glaucomatous mice are also capable of entering the retina with an intact blood-retinal barrier, although probably at a much slower rate or under certain conditions.

Presently, it is unclear which subsets of T cells serve as effector cells or act as initiators of glaucomatous neurodegeneration. T$_H$ cells are characterized by different cytokine profiles, which are key contributors to immune diseases and commonly used to

define the functional subsets of T cells[35]. Thus, in the present study, we investigated cytokine secretion profiles of CD4+ T cells, including IFN-γ (T$_{H1}$), IL-4 (T$_{H2}$), IL-17 (T$_{H17}$), and TGF-β (T-cell subsets with regulatory functions). We found that T cells from glaucomatous retina secreted IFN-γ, suggesting they are T$_{H1}$ cells, which are known to mediate inflammation and DTH reactions. T$_{H2}$ cells predominantly secrete IL-4 and stimulate the proliferation of B cells and production of antibodies; T$_{H17}$ cells produce proinflammatory cytokine IL-17 and play a major role in autoimmunity; Treg and those with a regulatory function, in contrast, are important for the control of immune responses to self-antigens, preventing autoimmunity and maintaining self-tolerance. While our data suggest that T$_{H1}$ cells are the predominant subset infiltrating the glaucomatous retina, it is increasingly recognized that "one pathogenic T$_H$ cell" and "one cytokine" does not fit the bill of autoimmune pathology anymore. Involvement of multiple different effector T-cell subsets in experimental autoimmune encephalomyelitis, via overlapping or distinct mechanisms, has been documented. Similar mechanisms can also be involved in glaucomatous neurodegeneration considering that IOP elevation induced activation of all four subsets of CD4+ T cells in the spleen and draining LN. Autoimmune responses in the retina usually start with the activation of microglia, which can function as antigen-presenting cells to capture and present antigens[36,37], such as HSPs. This in turn triggers HSP-specific T-cell responses in the draining LN and retinal production of inflammatory cytokines, such as tumor necrosis factor-α and IL-1β, that can continue after the IOP has returned to a normal range. These cytokines are known to weaken the blood-retinal barrier, allowing or facilitating T-cell infiltration into the retina[38,39]. Disruption of the blood-retina barrier has been reported in patients with POAG and in DBA/2J mice, correlating with the IOP elevation[40]. While we did not detect apparent vessel leakage in MB-induced glaucomatous mice using fluorescein angiography, we did observe T-cell infiltration in the retina of MB-injected eyes. Activated CD4+ T cells can cause neuronal damage directly as seen in multiple sclerosis or by activating other immune cell types, such as microglia, through secreted cytokines such as IFN-γ. Localized cytokine induction and microglia activation have likely limited the neural damage to the eye rather then spreading the damage to other sites of the central nervous system (CNS). Future examination of cytokine production and T-cell behavior in the retina of GF mice with elevated IOP will further help elucidate the cellular and molecular pathways underlying T-cell-mediated neurodegeneration to develop therapies. Although these questions remain to be addressed, our current study provides compelling evidence for commensal microbial flora-induced HSP-specific CD4+ T cells in the pathogenesis of chronic neurodegeneration in the eye.

## Methods

**Mice**. Adult (10–16 weeks old) male and female C57BL/6J (B6), $Rag1^{-/-}$, $TCR\beta^{-/-}$, $Igh6^{-/-}$, and GFP transgenic mice (all on a B6 background) and 3- to 12-month-old DBA/2J mice were purchased from the Jackson Laboratory. Mice of both sexes were evenly randomized into control and experimental groups. All experimental procedures and use of animals were approved and monitored by the Institute's Animal Care Committee and conform to the standards of the National Institute of Health and the Association for Research in Vision and Ophthalmology. GF and ASF colonized Swiss Webster mice of both sexes (10–16 weeks old) were originally purchased from Taconic Biosciences (Germantown, NY) and maintained at MIT in a genotobiotic core. DBA/2J mice were re-derived by embryo transfer into the GF health status using GF Swiss Webster recipient mothers. These mice were maintained in sterile plastic film isolators in open-top polycarbonate cages on autoclaved hardwood bedding and fed autoclaved water and diet (ProLab 2000, Purina Mills, St. Louis, MO) ad libitum. Macroenvironmental conditions included a 14:10 light/dark cycle and temperature maintenance at 68 ± 2 °F. The sterile plastic film isolators were confirmed to be maintained free of all aerobic and anaerobic bacteria, and mold by bimonthly microbiologic monitoring of feces from each cage mixed with drinking water, interior swabs of the isolator walls, and a

sampling of wet food left exposed to trap molds. These composite samples were screened for contaminants using aerobic and anaerobic culture, PCR, and fecal Gram stains. Control SPF mice (free of all murine bacterial pathogens, adventitious viruses, and parasites) were maintained under barrier conditions in standard microisolator cages under similar environmental conditions, including the same autoclaved diet and reverse osmosis-purified water. The ASF was originally developed for colonizing GF rodents with a defined microbiota, consisting of eight murine bacterial species: two aerotolerant lactobacillus strains, ASF360 and ASF361; two *Clostridium* sp. strains, ASF356 and ASF502; *Eubacterium* sp. strain ASF492; *Bacteroides* sp. strain ASF519; low-GpC-content Gram-positive bacterial strain ASF500; and strain *Mucispirillum schaedleri* strain ASF457[22,41]. A cohort of DBA/2J and Swiss Webster mice was maintained under SPF conditions to serve as controls. Unless specified, 6–8 mice were used per group in all experiments.

**Induction of IOP elevation in mice**. IOP was induced in adult mice (10–12 weeks old) as described previously[12]. Briefly, mice were anesthetized and supplemented with topical proparacaine HCl (0.5%; Bausch & Lomb Incorporated, Tampa, FL). Elevation of IOP was induced unilaterally in adult mice by an anterior chamber injection of polystyrene MBs having a uniform diameter of 15 μm (Invitrogen), which were re-suspended in sterile phosphate-buffered saline (PBS) at a final concentration of $5.0 \times 10^6$ MB/ml. The control group received an injection of sterile saline (2 μl) into the anterior chamber. Briefly, the right cornea was gently punctured near the center using a 30-gauge needle to generate an easy entry for injection. Following this entry wound, 2 μl of MB were injected into the right anterior chamber using a glass micropipette connected with a Hamilton syringe. Polystyrene MBs are widely used in the clinic and have proven to be safe without inducing significant levels of inflammation[42,43]. No clinical signs of anterior segment inflammation, including corneal ectasia or neovascularization, inflammatory cell infiltration into the anterior chamber, or synechiae and fibrosis in the angle area, were observed. The morphology of the anterior chamber and Schlemm's canal appeared normal in MB-injected mice, consistent with previous reports[44–46]. In all experimental groups, IOP was measured every other day in both eyes using a TonoLab tonometer (Colonial Medical Supply) as previously described[12]. The measurement of IOP was conducted consistently at the same time in the morning. Mice were anesthetized by isoflurane inhalation (2–4%; Webster Veterinary, Sterling, MA) delivered in 100% oxygen with a precision vaporizer.

Elevation of IOP in adult GF and ASF mice (16–20 weeks old) was performed in a sterile laminar flow hood under aseptic conditions. All equipment was gas-sterilized with ethylene oxide before entering the hood. Sterile drug vials and bottles were only used once, and discarded after the end of each experiment. IOP was induced unilaterally in the right eye, by anterior chamber injection of MB as described; control animals received an injection of sterile saline. For MB injection and periodic measurement of IOP, GF and ASF mice were exited from their home isolators in a sealed sterile container and transported to a Clidox (Pharmacal, Naugatuck, CT) sterilized hood. All manipulations of the mice were performed in the hood using strict aseptic technique. The mice were returned to separate, dedicated sterile isolators that were monitored closely for inadvertent contamination. Since retrieving mice out of the sterile isolators and any intervention on them increases the risk of contamination, IOP was measured shortly before anterior chamber injection, and subsequently on only 1, 2, and 8 weeks post injection. IOPs of GF DBA/2J mice were measured at 3, 8, 10, and 12 months of age.

**Immunohistochemistry and quantification of RGC and axon loss**. For immunohistochemistry, mice were sacrificed and perfused with saline for 5 min to remove all blood cells from their vessels followed by 4% paraformaldehyde for 15 min, and eyes and optic nerves were dissected. Retinal flat-mounts or transverse sections were incubated with a primary antibody against CD3, CD4, CD11b, Tuj1, Brn3a, and RECA1 (all from Invitrogen) followed by an Alexa Fluor 488-conjugated secondary antibody (Invitrogen). Quantification of RGC axon loss in optic nerve sections was as described[47], with minor modifications. All of the quantification procedures were routinely conducted under a masked fashion. Briefly, the optic nerve cross sections were traced using NIH Image J software, and the total area of each cross section was measured. For axon counts, five rectangular regions (64 μm × 85 μm) covering the entire optic nerve cross areas were photographed at ×1000 magnification (Nikon Eclipse E800), and four rectangular regions (30 μm × 30 μm) were cropped from each of the five rectangular areas. Axons were identified by the microbules and axolemma surrounded by the electron-dense myelin sheath, and all axons in the 20 selected regions, covering approximately 50% of the total optic nerve area, were counted. The percentage of axon loss was determined by dividing the axonal density calculated from the optic nerve with high IOP to that of the contralateral control optic nerve of the same mouse in B6 mice, or to that of 3-month-old (normal IOP) DBA/2J mice. RGC loss was assessed quantitatively in retinal flat-mounts using a standard protocol described previously[12] with minor modifications. In brief, retinal flat-mounts were incubated with a primary antibody against a RGC marker, β-III-tubulin[48] (Tuj1; Sigma-Aldrich, St.), followed by a Cy3-conjugated secondary antibody. Immuno-positive cells in 24 areas from each retina (×400; 12 areas taken from the peripheral retina, 8 from the intermediate regions, and 4 from the central retina), each covering 0.09 mm², were counted[12]. The numbers were averaged to calculate the RGC densities,

and the percentage of RGC loss was determined by dividing the RGC density calculated from the retina with high IOP by that of the contralateral control retina of the same mouse in B6 mice or by that of 3-month-old (normal IOP) DBA/2J mice.

**Isolation and adoptive transfer of CD4+ T cells**. Single-cell suspensions were prepared from spleens in RPMI-1640 media (Sigma) containing 10% fetal bovine serum (FBS), 1% penicillin/streptomycin, and 1% L-glutamine after lysis of red blood cells with red blood cell lysis buffer (Sigma). CD4+ T cells were purified using an auto MACS Separator and a CD4+ T Cell Isolation Kit (MiltenyiBiotec) according to the manufacturer's protocol. Briefly, CD4+ T cells were negatively selected from splenocytes of HSP27-immunized mice, or from mice with high IOP, by depletion with a mixture of lineage-specific biotin-conjugated antibodies against CD8 (Ly-2), CD11b (Mac-1), CD45R (B220), CD49b (DX5), Ter-119, and anti-biotin MBs. The procedure yielded purity higher than 90% CD4+ T cells, as assessed by flow cytometry. These cells ($2 \times 10^8$ cells/mouse in 200 µl PBS) were adoptively transferred into recipient mice via tail vein injection. Control animals received the same numbers of CD4+ T cells, isolated from mice with normal IOP or from OVA-immunized mice.

**ELISA and ELISPOT Assays**. Sera were collected from mice injected with MB or saline. Ninety-six-well plates (Nunc) were pre-coated with recombinant human HSP27 or HSP60 protein (1 µg/ml) blocked with 10% normal goat serum, followed by addition of diluted serum samples (1:10) and anti-HSP27 or anti-HSP60 antibody (1:10 as positive controls) and incubation for 2 h at room temperature. Horseradish peroxidase-conjugated anti-mouse (or human) IgG (1:1,000) were added and incubated for 45 min at room temperature. TMP substrate (Sigma) was added and optical density was measured at excitation wavelength 405 nm using XFlour4 software. Each sample was processed and analyzed in triplicate.

A mouse IFN-γ ELISPOT assay (eBioscience) was used to determine the frequency of IFNγ-producing T cells in response to HSP27 or HSP60. ELISPOT plates (Multiscreen-MAIPS4510), pre-coated with 100 µl/well of capture antibody, were blocked with 200 µl/well of complete RPMI-1640 medium. Mouse splenocytes ($1 \times 10^6$ cells/well) were added and incubated with antigens, including HSP27 or HSP60 (Sigma-Aldrich), E. coli HSP24 or HSP60 (EnzoLifesciences), IRBP, and MBP (Invitrogen), at a final concentration of 10 µg/ml, for 48 h. Cell cultures incubated alone were used as controls. Results are shown as the mean number of antigen-specific spot forming cells after background subtraction from the control wells containing no antigen.

**Flow cytometry assays**. The mouse cervical LNs, spleen, or retina were freshly isolated. LNs and spleens were dissociated using forceps; the cell mixtures were filtered through a 70 µm nylon mesh. For the neuroretina, two to three retinas were combined for each test and dissociated using Papain Dissociation system (Worthington Biochemical Product). Dissociated cells taken from the cervical LNs, spleen, or retina were immunolabeled with fluorescein isothiocyanate (Biolegend)- or PerCP/Cy5.5 (BD Bioscience)-conjugated anti-CD4 antibody. The cells were then permeabilized with permeabilizing/washing buffer (Biolegend), followed by immunolabeling for cytokines with Alexa Fluor 488 (BD Bioscience)- or phycoerythrin-conjugated primary antibodies against IFN-γ, IL-4, IL-17, or TGF-β (BioLegend) for 30 min at 4 °C in dark. Corresponding isotype antibodies were used as controls. Data were collected with BD LSR II Flow Cytometer as 20,000–30,000 events and analyzed by FlowJo (Tree Star).

**Preparation of human blood samples and T-cell assays**. Patients (40–60 years) who had been diagnosed with POAG with unambiguous clinical evidence of pathological "cupping" of the optic nerve head and documentation of visual field loss, were recruited for this study. Recruited control subjects showed no evidence of optic nerve or CNS damage from any cause, and were found to have no significant visual or neurological disorder. In addition, patients with retinal detachment as well as those suffering from skin injuries were examined to exclude the possibility that the increase in HSP27- and Hsp60-responsive T cells in POAG was a result of stress responses to inflammation or disease conditions independent of POAG. All participants were provided with informed consent to participate in the relevant research resources. The study was approved by the Institutional Review Board (IRB) of the Massachusetts Eye and Ear. Sixteen milliliters of venous blood was drawn from each volunteer into a vacutainer CPT tube (Becton Dickinson) with sodium citrate and processed according to the manufacturer's instructions. The peripheral blood mononuclear cells were re-suspended at $1.0 \times 10^7$ cells/ml in RPMI containing 20% heat-inactivated FBS. ELISPOT and ELISA assays were performed as described above. The measurements were performed in a masked fashion.

**Delayed-type hypersensitivity assay**. The thickness of the mouse ear was measured using a micrometer prior to antigen stimulation. The dorsal side of the ear was injected intradermally with 10 µl of human recombinant HSP27 (1 µg/µl; Enzo Life Science) or with a control antigen—either MBP or IRBP (1 µg/µl; Invitrogen). The thickness of the injected ear was measured again after 24 and 48 h, respectively, and change in thickness was calculated. We did not find significant

differences between the results obtained at 24 and 48 h; thus, only results obtained at 24 h were reported.

**HSP27 inoculation**. To immunize mice, 50 µl of human recombinant HSP27 (50 µg; Enzo Life Science) was emulsified with 50 µl incomplete Freund's adjuvant emulsion and injected subcutaneously in adult B6 mice. Two to three weeks later, immune responses to HSP27 were analyzed by DTH and ELISPOT assays.

**Statistical analysis**. Based on the 90% statistical power analysis done with our previous data in glaucoma studies, six mice per group were shown to yield over 95% significance, two-sided test[12,49–51]. This is also the group size commonly used in other reports seeking statistical significance in rodent models of experimental glaucoma[52,53]. Thus, most of our experiments used a group size of six to eight animals, unless otherwise indicated. Scatter and bar plots are presented as mean ± s.e.m., and a paired analysis of variance was used for statistical analysis.

**Data availability**. The authors declare that the data supporting the findings of this study are available within the article and its Supplementary Information files, or are available upon reasonable requests to the authors.

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

## Acknowledgements

We thank Randy Huang of the Schepens Eye Research Institute Flow Cytometry Core Facility for assistant in data analysis, Eric Zhao for schematic graphic design, and MIT technicians Carlos Umana, Oscar Acevedo, and Allan Discua for performing the embryo transfer and maintaining the GF and ASF mice. This work was supported by grants from the National Institutes of Health (EY025913, EY027067, and EY025259; NS038253; and AI69208), Lion's Foundation, Miriam and Sheldon Adelson Medical Research Foundation, the National Nature Science Foundation of China (81200683 and 20120162), the Ivan R. Cottrell Professorship and Research Fund, Oogfonds/StichtingGlaucoomfonds/SNOO/LOOF/Nelly Reef Fund/Prins Bernhard Cultuur Fonds, the Koch Institute Support (core) Grant P30-CA14051 from the National Cancer Institute, and the NEI Core Grant for Vision Research (P30 EY03790).

## Author contributions

D.F.C. and J.C. designed and conceived the study, and H.C., K.-S.C., T.H.K.V., C.-H.S., M.K., G.C., R.M., H.Y., Y.L., and L.Y. performed the experiments and data collection. R.M. and J.S.-S. performed DTH assays in B6 mice, and M.L.M., A.F., and K.L. provided human blood samples. N.P.B.A., J.K.Y.T., C.H.E.M., and C.J.W. provided HSP27 transgenic mouse and performed culture studies with HSP27 transgenic mice. M.T.W. and J.G.F. developed/re-derived germ-free DBA/2J mice and ASF mice, and M.J.J., J.C., and D.F.C. analyzed data, and wrote and revised the manuscript.

## Additional information

**Competing interests:** H.C., K.-S.C., D.F.C., and J.C. are co-inventors on a pending patent application for targeting T cells and microbiome for optic neuropathy therapy. The other authors declare no competing interests.

