## [Peer Review File · Nature Communications]

Reviewers' comments:

Reviewer #1 (neuro-immune crosstalk)(Remarks to the Author):

This manuscript attributes a negative role to T cells in two animal models of glaucoma, IOP-induced and a spontaneous developed in mice.

While the results are intriguing, there are some key issues that needed to be addressed before one can conclude that the T cells with reactivity against commensal microflora are responsible for the escalation in glaucoma.

1. The authors demonstrated that the number of neurons that survive 2 weeks after insult in WT animals and in Rag1^{-/-} mice is the same, but 8 weeks later more neurons are lost in the WT. Based on these results, the authors concluded that T cells are negative players in coping with IOP-induced insult. To justify this conclusion, the authors need to include more controls: They should test whether BM transplantation from WT but not from Rag1^{-/-} reverses the outcome in Rag1^{-/-} mice. Transfer of a specific subpopulation of T cells to immunocompromised animals (Rag1^{-/-}) mice might create a different effect than that of the presence of the same cells in immune competent mice, due to homeostasis-driven proliferation that occur in Rag1^{-/-} mice.

2. Transfer of T cells specific for human MBP is not a relevant control (Figure 3).

3. The authors should test whether the effect is due to effector or regulatory T cells; transferring the total T cell population is not informative enough.

4. In Figure 4 the authors claimed that they isolated T cells from the retina and expanded them with specific HSP27 antigen. How many T cells could be isolated? Does the expanded population also include regulatory T cells?

5. The lack of neuronal death in GF DBA1 mice cannot be interpreted to prove the role of T cells. The authors should have crossed DBA1 with Rag1^{-/-} mice to demonstrate this effect.

Reviewer #2 (glaucoma, autoimmune)(Remarks to the Author):

This manuscript described for the first time that a transient elevation of IOP by microbeads injection can induce CD3⁺ cell infiltration into the retina, and only with IOP elevation but without T cells the neurodegeneration could not develop, indicating the essential roles of T cells, microbiota and several specific HSPs in glaucoma pathogenesis. Interestingly, according to the results here, the presence of microbiota is an essential condition for the neurodegeneration of RNFL and RGCs, which seems to be mutually exclusive against the other hypotheses such as mechanical pressuring or trans-lamina-cribosa pressure difference or oxidative stress, etc.

In summary, this paper shows a variety of experiments including different groups of animals and human samples as well. In our opinion, the manuscripts structure should be reorganized to yield a better overview. Also, the separation of figures in main figures and suppl figures is quite misleading. Additionally, some figures do not show the indicated data, and even worse, some figures show the exact same data. I recommend a major revision to improve the structure of the text as well as the clarity of the figures.

General Comments:

Please include a detailed list of the number of animals, strain and sex as well as age. "6 to 8 mice per group" is not scientific statement. The description of animal numbers in the figure legends is misleading.

Please improve the Material part by including the suppliers of chemicals and instrument and their site of location.

Specific Comments:

Line 111: Supp Fig 1 b does not show triple staining as indicated in the text, but cd11b staining as indicated in the figure legends. Arrows are hardly visible.

Line 114: Fig 1 c: Specify T cells as CD3-positive cells, same procedure for fig. 1 d.

Line 115, Figure 1e: For the FACS part, why only the CD4+ cells were chosen? And how many events were recorded for each FACS sample? The method part of CD4 flow cytometry is absent.

Line 118: Supp Fig 1 d: It is impossible to distinguish red and purple signals in the merged image. Please provide the single channels and a merged image. Scale bar is missing.

Line 120: If you include different concentrations of MB, please describe the results in detail. Please clearly label the low concentration with detailed information. Include suppl Fig 1 e in the main fig 1.

Line 123: Figure describe DBA mice as 8-10 months old. Text says mice were 8 months old. Please clearly indicate how many animals were included with the exact age and standard deviation.

Line 128: It has been shown various times that Tuj1 and Brn3a are suitable markers for RGCs. Since there is no new scientific knowledge included in fig 1 h-I, we recommend to exclude the data from this manuscript.

Line 131: Clearly indicate which labeling was used for analysis of RGC data (fig. 1 h and j).

Line 138: The percental RGC loss/ axonal loss only presents the previous data after calculation.

We recommend to exclude this data and to add IOP data on the secondary y-axis to fig 1 h/i.

Line 140: I recommend to display CD3, CD4, CD11b data in a compact subfigure in the main figure 1 to yield a better overview.

Line 151: How would you explain that neurodegeneration is hampered in TCR β mice in RGCs but not in axons?

Line 157: Why only CD4+ cells were transferred other than CD3+ cells or just memory T cells, such as CD4+CD27+CD28+ cells, more specifically?

Line 158: It is confusing here: was the MB injection simultaneous for these two strains? Or the MB injection in the Rag1 $^{-/-}$ was 14 days earlier than in the B6 mice?

Line 161: Fig 2 d: Please show CD4/TUJ1/DAPI images for MB, U and S group. A quantification of CD4 positive cells per group is missing.

Line 163: Why does the uninjected mice (U) show 15% loss in RGC and axons?

Line 166: Supp Fig 2 b-e, indicate as percental loss to facilitate comparability to figure 2 f and g.

Line 168: Replace "diseased" by "conditioned"

Line 174: protein levels or antibody levels? It is not clear why they focus on protein levels in the later part when autoantibodies are differently regulated in patients and animal models.

Line 177: How can you prove that HSP60 was found in other layer? Retinal traverse sections would be more useful in this approach to identify colocalization in specific layers of the retina.

Line 179: Indicate that this data is about antibodies. Why did the authors exclude the 8 weeks post injection time point?

Line 185: Fig 3 d: μ m instead of mm

Line 187: Supp Fig 3 a, Rag $^{-/-}$ wit saline injection is missing. Quantification of CD4-positive cells is missing.

Line 218: Include Supp fig 3 d and e to fig 4.

Line 246: HSP27 staining quantification is missing. CTRL group images are not convincing.

Line 232, 352: Blood-retinal barrier.

Line 261: Fig 5 g,h: RGC loss and Axon loss seem to be the VERY same in all groups. This is hard to believe. Please make sure to analyze this data again in detail.

Line 393: Bausch.

Line 396, 435, 436, 437 etc: "x" instead of "x".

Line 505: It would be better if the interval between two measurements was longer, for the interval of 24 hours sometimes might be too short to see delayed type hypersensitivity. The inflammatory reaction to antigen peaks within 48 hours post exposure.

Figure 4a & b: The data seem to be not corresponding in these two figures {0.17%/(100%-99%-0.38%)} vs. 0.40%/(100%-98%-1.01%)} . And the gate should be better designed as the

majority of the total events (99% & 98% here) fell in the left-lower-quadrant. Plus, it would be more persuasive to show the isotype controls of these samples.
Supplementary Fig. 1h: "marge" or maybe "merge"?

Reviewer #3 (glaucoma, IOP model)(Remarks to the Author):

This is a manuscript reporting data that suggest that a relatively short term IOP elevation (3 weeks) induces an acute neurodegeneration as well as chronic neurodegeneration which is mediated by T cell infiltration of the retina. The authors determined that at least some of these T cells are reactive to HSPs. Given the similarity between bacterial and human HSPs the authors then proceeded to show that in animals raised under germ-free condition the chronic neurodegeneration phase does not occur. Finally the authors report that patients with glaucoma have elevated T-cell responsiveness to HSPs compared with control groups. The experimental work performed is well described and without deficiencies in the experimental design and performance. Controls are rigorous and experiments are robust. Use of two separate glaucoma models significantly strengthens the manuscript and some of the conclusions reached. Inclusion of multiple controls also adds confidence.

A few additional pieces of data and a better discussion would further strengthen this manuscript:

1. It would be useful if the authors could provide an indication of the spatial distribution of T-cells infiltrating the retina after IOP elevation. Is it diffuse or localized in clusters? Is it more prominent in one or more quadrants? Is it more common in center or periphery?

2. In Figure 3 quite a bit of immunostaining for HSP27 is seen extracellularly. Why is that?

3. Discussion is appropriate but fails to include two very relevant papers:

One by Gramlich et al (*Acta Neuropathol Commun.* 2015 Sep 15; 3:56), also reports involvement of T-cells in glaucoma neurodegeneration. However, contrary to what the authors report here, the previous paper documents that neurodegeneration can be caused by adoptive transfer of T-cells even in the absence of IOP elevation. The authors of the present manuscript should discuss the possible reasons for this discrepancy.

The other is the paper by Astafurov et al (*PLoS One.* 2014 Sep 2; 9(9):e104416) who reported that peripherally administered LPS can exacerbate neurodegeneration in glaucoma. Again the authors should discuss how their work relates to those findings.

In addition discussion fails to comment on a number of unanswered questions:

a. Why does neurodegeneration in the retina and ON continue after IOP normalizes? If activated T-cells can access these tissues without IOP elevation, then why don't they do so in the rest of the CNS?

b. Do other conditions that involve HSP upregulation and tissue damage (which presumably would also sensitize T-cells) cause neurodegeneration in the retina and ON?

c. Who are the effector cells that cause neurodegeneration? Are they T-cells? B-cells? Microglia? Monocytes? Astrocytes? Microglia numbers are provided by the authors but despite a decline in microglial activation after normalization of IOP damage continues. Why?

d. Although one can envision how T-cells "see" HSPs when there is cell/tissue damage it is unclear how they would "see" HSPs in an intact RGC.

Obviously many of these questions cannot be answered with the current level of knowledge but it would be important for the authors to at least speculate on possible mechanisms.

Nevertheless this is a very important study and the amount of work done is exceptional.

Reviewer #1:

“1. The authors demonstrated that the number of neurons that survive 2 weeks after insult in WT animals and in Rag1^{-/-} mice is the same, but 8 weeks later more neurons are lost in the WT. Based on these results, the authors concluded that T cells are negative players in coping with IOP-induced insult. To justify this conclusion, the authors need to include more controls: They should test whether BM transplantation from WT but not from Rag1^{-/-} reverses the outcome in Rag^{-/-} mice. Transfer of a specific subpopulation of T cells to immunocompromised animals (Rag1^{-/-}) mice might create a different effect than that of the presence of the same cells in immune competent mice, due to homeostasis-driven proliferation that occur in Rag1^{-/-} mice.”

Response: We appreciate the reviewer’s careful consideration and comment. We would like to point out that BM transplantation to Rag1^{-/-} mice may not eliminate the homeostasis-driven proliferation and differentiation as newly generated T cells from thymuses of adult Rag1^{-/-} recipient mice following BM transplantation would also undergo homeostasis-driven proliferation. Plus, it has been shown that Rag1^{-/-} recipients of BM cells exhibit a low level reconstitution of T cells due to low cellularity of their thymus (van Til NP et al, J Allergy Clin Immunol; 2014); thus, we believe that BM transplantation to Rag1^{-/-} mice would be unable to answer the reviewer’s question.

We consider that the best approach to address the reviewer’s concern which T cell transfer to Rag1^{-/-} mice may “create a different effect than that of the presence of the same cells in immune competent mice,” may be to perform T cell transfer from a glaucomatous donor to WT recipients. As mentioned by Reviewer

#3, this study has been done by another group (Gramlich et al, 2015 Acta Neuropathol Commun), using T cells isolated from at least 2 mouse lines that show an elevated IOP due to genetic mutations to wild-type recipients. Their results are in agreement with our finding presented in this paper, which support a role for T cell-mediated responses in glaucomatous neural damage. This study and its citation have now been included (Ref. #34; lines 370 – 372).

“2. Transfer of T cells specific for human MBP is not a relevant control (Figure 3).”

Response: We apologize for a potential confusion here. In our study, human MBP was used as a control antigen for culture stimulation in ELISPOT assays, in which we demonstrated that ocular hypertension induced a T cell response specific to HSPs, but not to MBP or IRBP (Fig. 3d). We agree with the reviewer that transfer of T cells specific for human MBP is not a relevant control. Instead, we used T cells isolated from naïve or saline-injected mice with normal IOP as the controls (Fig. 2). We have revised the text and clarified this potential confusion (lines 157 – 158).

“3. The authors should test whether the effect is due to effector or regulatory T cells; transferring the total T cell population is not informative enough.”

Response: To test potential effector cells, we have analyzed cytokine expression by CD4⁺ T cells from spleens and eye draining LNs. Our data show that elevated IOP induced CD4⁺ T cell expression of IFN- γ (T_H1), IL-4 (T_H2), IL-17 (T_H17) and TGF- β (Treg) without apparently biasing toward any specific subset (Supplementary Fig. 2b,c and lines 146 – 152). These results support the induction of CD4⁺ T cell responses by elevated IOP. While it is more informative to transfer T cell subsets, it is technically more challenging as enough subsets of T cells have to be sorted from diseased mice without using intracellular cytokine staining. Moreover, as it has been reported in other disease models, it is very likely that not only a single subpopulation, but multiple different T cell subsets are involved in mediating the autoimmune process, as we have mentioned in the Discussion section (lines 377 – 381). We are certainly interested in examining this issue in the future.

“4. In Figure 4 the authors claimed that they isolated T cells from the retina and expanded them with specific HSP27 antigen. How many T cells could be isolated? Does the expanded population also include regulatory T cells?”

Response: We have now added that ~1,000 CD4⁺ T cells were detected per glaucomatous mouse retina by flow cytometry (Fig. 1e; lines 216 – 217). Only INF- γ ⁺CD4⁺ double-labeled T cell population was detected, indicating that they are T_H1 type cells; we were unable to detect the other subsets of CD4⁺ T cells (T_H17, T_H2, and Treg) in the expanded cell population, likely because they constituted a very small proportion of the infiltrating T cells. This has now been clarified (lines 217 – 220).

“5. The lack of neuronal death in GF DBA1 mice cannot be interpreted to prove the role of T cells. The authors should have crossed DBA1 with Rag1^{-/-} mice to demonstrate this effect.”

Response: The genetic mutation(s) in DBA/2J mice that lead to IOP elevation and the glaucoma phenotype have (has) not been identified. Transfer of *Tyrp1*^b and *Gpnmb*^{R150X} mutations identified in DBA/2J mice to B6 mice (by crossing DBA/2J with B6 mice) induced a DBA/2J-like iris disease without the development of high IOP or glaucomatous neural damage. To carry out the study suggested by the reviewer, one would need to knockout *Rag1* in DBA/2J mice or breed *Rag1*^{-/-} onto the DBA/2J background (ideally for 20 generations). Even after doing all this and if positive results were obtained, one may still question whether this outcome proves the role of T cells or is a result of a varied genetic background.

By re-deriving germ-free DBA/2J mice, our result provides the compelling evidence that elevated IOP in glaucoma does not directly attribute to progressive neurodegeneration; it is the subsequent event known to involve the immune system and T cell responses that plays a key role in this process. Although the

lack of neuronal damage in GF DBA/2J mice cannot be interpreted as a direct prove for the role of T cells by itself, our results together demonstrate a critical role for T cells in glaucomatous neurodegeneration and the important role of how microflora contributes to it. We appreciate very much the reviewer's point and have now added a paragraph in the Discussion section to clarify this notion (lines 318 – 324).

Reviewer #2:

“In our opinion, the manuscripts structure should be reorganized to yield a better overview. Also, the separation of figures in main figures and suppl figures is quite misleading. Additionally, some figures do not show the indicated data, and even worse, some figures show the exact same data.”

Response: We have now carefully gone through the entire manuscripts and rearranged the structure as well as some of the main and supplementary figures accordingly to eliminate duplications and give a better flow of the results.

General Comments:

“Please include a detailed list of the number of animals, strain and sex as well as age. “6 to 8 mice per group” is not scientific statement. The description of animal numbers in the figure legends is misleading. Please improve the Material part by including the suppliers of chemicals and instrument and their site of location.”

Response: The numbers of animals of each experiment were updated, and more details on strains, sexes, and ages have now been included (lines 397 – 403); suppliers of chemicals and instruments and their site of location have now been added to the Material and Methods section.

Specific Comments:

Line 111: Supp Fig 1 b does not show triple staining as indicated in the text, but cd11b staining as indicated in the figure legends. Arrows are hardly visible.

Response: Supplementary Fig. 1b was misplaced to 1d. This error has now been corrected. The arrows in Supplementary Fig. 1b have been enlarged to make them more visible.

Line 114: Fig 1 c: Specify T cells as CD3-positive cells, same procedure for fig. 1 d.

Response: We have now specified that CD3 is a general T cell marker (line 93) and clarified in Fig. 1d legend (line 733) that CD4⁺ detects a T cell marker.

Line 115, Figure 1e: For the FACS part, why only the CD4+ cells were chosen? And how many events were recorded for each FACS sample? The method part of CD4 flow cytometry is absent.

Response: The procedure of flow cytometry analysis has now been added to the Methods section, and over 20,000 events were recorded from each sample (lines 516 – 527). We have also clarified that to define the subpopulations of infiltrating T cells, we performed immunolabeling and flow cytometry analysis for CD4 and CD8 T cells, and only CD4⁺ T cells were detected (lines 98 – 102). Hence, our subsequent studies have focused on CD4⁺ cells.

Line 118: Supp Fig 1 d: It is impossible to distinguish red and purple signals in the merged image. Please provide the single channels and a merged image. Scale bar is missing.

Response: The single channels and the merged images, with a scale bar, have now been added (Supplementary Fig. 1b).

Line 120: If you include different concentrations of MB, please describe the results in detail. Please clearly label the low concentration with detailed information. Include suppl Fig 1 e in the main fig 1.

Response: We have now included the different concentrations of MB and described the results in further detail in both the main text (lines 105 – 110) and the figure legends for Fig. 1h (lines 743 – 744) and Supplementary Fig. 1c (line 869).

Line 123: Figure describes DBA mice as 8-10 months old. Text says mice were 8 months old. Please clearly indicate how many animals were included with the exact age and standard deviation.

Response: We have now clarified in both Fig. c and Supplementary Fig. 1 legend that for 3 (n=6) and 8 (n=8) months old DBA/2J mice (line 732 and 871). The IOP and RGC quantification data taken from 10 month old DBA/2J mice included in the previous Supplementary Fig. 1f,g have now been removed. The values and S.E.M. have been adapted.

Line 128: It has been shown various times that Tuj1 and Brn3a are suitable markers for RGCs. Since there is no new scientific knowledge included in fig 1 h-I, we recommend to exclude the data from this manuscript.

Response: Original Fig. 1h,i have been removed per reviewer's suggestion.

Line 131: Clearly indicate which labeling was used for analysis of RGC data (fig. 1 h and j).

Response: We have now clarified in both the main text (lines 117 – 118) and the legend to Fig. 1h (line 750), that Tuj1 immunolabeling was used primarily to quantify RGC loss.

Line 138: The percental RGC loss/ axonal loss only presents the previous data after calculation. We recommend to exclude this data and to add IOP data on the secondary y-axis to fig 1 h/i.

Response: To be consistent with the data presentation in the rest of the manuscript, we have now deleted original Fig. 1h,i which represent the RGC and axon densities while moved the original Fig. 1j,k that showed the percentages of RGC and axonal loss to Fig. 1h,i. The IOP data has been added to the secondary y-axis of the current Fig. 1h,i as it is more visible in this presentation than when we had added them to the original Fig. 1h,i.

Line 140: I recommend to display CD3, CD4, CD11b data in a compact subfigure in the main figure 1 to yield a better overview.

Response: We appreciate the reviewer's comments and have now changed the subfigure accordingly.

Line 151: How would you explain that neurodegeneration is hampered in TCRβ mice in RGCs but not in axons?

Response: We have now clarified: "In $TCR\beta^{-/-}$ mice, which lack $CD4^{+} \alpha\beta T$ cells but are compensated by expansion of $\gamma\delta$ and NK T cells and B cells, no significant further loss of RGCs was detected between 2 and 8 weeks after MB injection; whereas, attenuated albeit significant loss of axons was observed at 8 weeks. The data suggest a primary role for $CD4^{+} \alpha\beta T$ cells, with a possible involvement of other immune cells, in glaucomatous neural damage." (lines 136 – 140).

Line 157: Why only CD4+ cells were transferred other than CD3+ cells or just memory T cells, such as CD4+CD27+CD28+ cells, more specifically?

Response: $CD4^{+}$ T cells were transferred because we noted that the majority of T cells which infiltrated the glaucomatous retinas were $CD4^{+}$. We have also analyzed cytokine expression by $CD4^{+}$ T cells from spleens

and eye draining LNs. Our data revealed that elevated IOP induced CD4⁺ T cell expression of IL-17 (T_H17), IFN- γ (T_H1), IL-4 (T_H2) and TGF- β (Treg) without apparently biasing toward any specific subset (Supplementary Fig. 2b,c; lines 146 – 152). These results support the induction of multiple subsets of CD4⁺ T cells by elevated IOP, and suggest that it may be most effective to transfer all, rather than a subset of CD4⁺ T cells. This was clarified (line 155).

To further clarify, we have also added in the Discussion section: “At this point, little is known regarding which subsets of T cells serve as effector cells or act as initiators of glaucomatous neurodegeneration. Involvement of multiple different effector T cell subsets in experimental autoimmune encephalomyelitis, via overlapping or distinct mechanisms, has been documented. Similar mechanisms can also be involved in glaucomatous neurodegeneration considering that IOP elevation induced activations of all four subsets of CD4⁺ T cells in the spleens and draining LN.” (lines 375 – 381). We are definitely interested in further examining the involvement of specific subsets of T cells in the future, including looking at memory T cells.

Line 158: It is confusing here: was the MB injection simultaneous for these two strains? Or the MB injection in the Rag1^{-/-} was 14 days earlier than in the B6 mice?

Response: We have now rewritten the sentence to clarify that Rag1^{-/-} mice received MB injection 14 days earlier, the same day as the donor mice received MB injections (line 159).

Line 161: Fig 2 d: Please show CD4/TUJ1/DAPI images for MB, U and S group. A quantification of CD4 positive cells per group is missing.

Response: Quantification of CD4⁺ T cells has now been added (Fig. 2f). Anti-CD4-labeled images of S and MB-injected groups and anti-CD4/Tuj1/DAPI triple-labeled image of the MB-injected group are included (Fig. 2d,e).

Line 163: Why does the uninjected mice (U) show 15% loss in RGC and axons?

Response: We have now clarified in the Fig. 2d-h legend that “U” represents the group of glaucomatous (MB-injected) Rag1^{-/-} recipient mice that received no CD4⁺ T cell transfer (uninjected or no T cell transfer). Moreover, we stated in the revised manuscript: “Two weeks after CD4⁺ T cell transfer (or 4 weeks after MB injection), glaucomatous Rag1^{-/-} recipients that had not received any injection of CD4⁺ T cells (U)... as expected, exhibited ~15% RGC and axon losses as a result of the elevated IOP-induced initial phase of neural damage (**Fig. 2g,h**). In contrast, MB injected glaucomatous Rag1^{-/-} recipients that had received a CD4⁺ T cell transfer from glaucomatous (MB-injected) B6 mice showed retinal T cell infiltration and a significant further increase in RGC and axon loss ...” (lines 160 – 167)

Line 166: Supp Fig 2 b-e, indicate as percental loss to facilitate comparability to figure 2 f and g.

Response:

Shown above are the bar charts converted from RGC and axon density comparisons to the percentage loss. We find these charts rather confusing as they tend to distract readers to the smaller than 2% differences

between groups rather than to focus on the little differences in RGC and axon densities between mice received control or diseased T cell/IgG transfers as shown in the original figure. Therefore, we decide not to convert the charts but present the converted bar charts above for the reviewer to see our point.

Line 168: Replace “diseased” by “conditioned”

Response: This has been changed.

Line 174: protein levels or antibody levels? It is not clear why they focus on protein levels in the later part when autoantibodies are differently regulated in patients and animal models.

Response: We have added “proteins” following “HSP27 and HSP60” in the original line 174 (current line 181) to clarify that these were protein levels. We have further clarified that “to define the autoantigens that stimulate T cell activation in glaucoma,... which may be expressed at a low level under the normal condition but are upregulated in RGCs in glaucoma”, we examined the levels of expression change of HSPs before and after IOP elevation (lines 176 – 181).

Line 177: How can you prove that HSP60 was found in other layer? Retinal traverse sections would be more useful in this approach to identify colocalization in specific layers of the retina.

Response: Yes, immunostaining for HSP27 and HSP60 was performed in retinal transverse sections, and these data have now been included (Supplementary Fig. 2h).

Line 179: Indicate that this data is about antibodies. Why did the authors exclude the 8 weeks post injection time point?

Response: It is now stated in both the text and figure legend that the data is about antibodies. The data of 8 weeks post injection was included but was mis-labeled for “4w” in the previous Fig. 3c. This has now been corrected, and we apologize for the error.

Line 185: Fig 3 d: μm instead of mm

Response: This has been corrected.

Line 187: Supp Fig 3a, Rag^{-/-} wit saline injection is missing. Quantification of CD4-positive cells is missing.

Response: As almost no T cells were detected in the retina of saline-injected Rag1^{-/-} mice, similar to what had been shown in the retinal flat-mount image of MB-injected Rag1^{-/-} mice in Fig. 3a, we do not feel that it is necessary to show two images with a black background. DTH responses are commonly quantified by measuring ear thickness, as we have shown in Fig. 3c. Counting infiltrating CD4⁺ T cells in ear sections is merely a duplication but a less used method for DTH response quantification compared to assessing ear thickness. However, we include the images of T cell immunolabeling taken from the ear sections of WT and Rag1^{-/-} mice as they present clearly visible differences.

Line 218: Include Supp fig 3 d and e to fig 4.

Response: Supplementary Fig. 3d,e have now been moved to Fig. 4e,f.

Line 246: HSP27 staining quantification is missing. CTRL group images are not convincing.

Response: The photomicrographs of HSP27 immunostaining in the normal IOP groups of SPF and GF mice have been replaced with better quality and more convincing images. The staining was performed for 3 mice per group, and we did not notice significant differences between the GF and SPF group before and after IOP elevation. This point has now been clarified (lines 256 – 259).

Line 232, 352: Blood-retinal barrier.

Response: These have been corrected.

Line 261: Fig 5 g,h: RGC loss and Axon loss seem to be the VERY same in all groups. This is hard to believe. Please make sure to analyze this data again in detail.

Response: We thanks for the reviewer's comment, and the bar chart of RGC loss was mistakenly put down twice. We have now replaced it with the correct bar chart of the axon loss data (Fig. 5h).

Line 393: Bausch.

Response: This has been corrected.

Line 396, 435, 436, 437 etc: "×" instead of "x".

Response: These are corrected.

Line 505: It would be better if the interval between two measurements was longer, for the interval of 24 hours sometimes might be too short to see delayed type hypersensitivity. The inflammatory reaction to antigen peaks within 48 hours post exposure.

Response: In fact, we assessed the reactions at both 24 and 48 hours and found similar results, so only the data obtained at 24 hr interval were reported. To reflect this point, we have now revised the paragraph to: "The thickness of the injected ear was measured again after 24 and 48 hours, respectively, and change in thickness was calculated. We did not find significant differences between the results obtained at 24 and 48 hours; thus, only results obtained at 24 hours were reported." (lines 548 – 551).

Figure 4a & b: The data seem to be not corresponding in these two figures {0.17%/(100%-99%-0.38%)} vs. 0.40%/(100%-98%-1.01%)}?. And the gate should be better designed as the majority of the total events (99% & 98% here) fell in the left-lower-quadrant. Plus, it would be more persuasive to show the isotype controls of these samples.

Response: Flow cytometry staining profile of the isotype control has now been included in Fig. 4b, and new anti-CD4 vs anti-IINF- γ -stained profiles from retinal cells of saline- and MB-injected mice that better represent the quantification data shown in Fig. 4a are provided. We have further clarified in Fig. 4 legend: "Shown are frequencies of INF- γ ⁺ cells in CD4⁺ gated cells (a)" (lines 806 – 807)

Supplementary Fig. 1h: "marge" or maybe "merge"?

Response: This has been corrected to "merge".

Reviewer #3:

1. It would be useful if the authors could provide an indication of the spatial distribution of T-cells infiltrating the retina after IOP elevation. Is it diffuse or localized in clusters? Is it more prominent in one or more quadrants? Is it more common in center or periphery?

Response: We have now added that infiltrated "Infiltrating T cells were noted at 2 weeks after MB-injection (**Fig. 1c**), scattered throughout the retina without apparent clustering or preference to any specific quadrant." (lines 95 – 97).

2. In Figure 3 quite a bit of immunostaining for HSP27 is seen extracellularly. Why is that?

Response: We have now clarified: “HSP27 upregulation was found primarily in the GCL (**Supplementary Fig. 2h**), both associated with RGC membranes and outside RGC bodies (**Fig. 3b**). This is consistent with the report that HSP27 is also upregulated in astrocytes following elevated IOP and be released extracellularly under stress^{18,19}.” (lines 183 – 186).

3. Discussion is appropriate but fails to include two very relevant papers:

One by Gramlich et al (Acta Neuropathol Commun. 2015 Sep 15;3:56), also reports involvement of T-cells in glaucoma neurodegeneration. However, contrary to what the authors report here, the previous paper documents that neurodegeneration can be caused by adoptive transfer of T-cells even in the absence of IOP elevation. The authors of the present manuscript should discuss the possible reasons for this discrepancy. The other is the paper by Astafurov et al (PLoS One. 2014 Sep 2;9(9):e104416) who reported that peripherally administered LPS can exacerbate neurodegeneration in glaucoma. Again the authors should discuss how their work relates to those findings.

Response: We appreciate the reviewer’s comments and have now added in the Discussion section: “induction of retinal damage by adoptive transfer of CD4⁺ T cells is associated with local injury/inflammation... This is seemingly supported by the observation that peripheral administration of LPS exacerbated neuron loss in glaucoma³³. Another report, however, showed that neurodegeneration could be induced in mice even with a normal IOP 8 weeks or longer after adoptive transfer of T cells isolated from genetic mouse models of glaucoma³⁴. It suggests that activated T cells of glaucomatous mice are capable of entering the retina with an intact blood-retinal barrier, although probably at a much slower rate or under certain conditions.” (lines 368 – 376)

In addition discussion fails to comment on a number of unanswered questions:

a. Why does neurodegeneration in the retina and ON continue after IOP normalizes? If activated T-cells can access these tissues without IOP elevation, then why don’t they do so in the rest of the CNS?

Response: We have now clarified further: “autoimmune responses in the retina start with activation of microglia, ...which leads to retinal production of inflammatory cytokines, such as TNF- α and IL-1 β , that can continue after IOP returns to a normal range. These cytokines weaken the blood-retinal barrier, allowing or facilitating more T cell infiltration into the retina^{37,38}... Localized cytokine induction and microglia activation likely has limited the neural damage to the eye rather spreading widely to other sites of the CNS.” (lines 382 – 392)

b. Do other conditions that involve HSP upregulation and tissue damage (which presumably would also sensitize T-cells) cause neurodegeneration in the retina and ON?

Response: We have now added: Other studies from our group have revealed that “T cell sensitization and progressive neurodegeneration were also noted in other conditions involving HSP upregulation, including retinal ischemia and optic nerve trauma (unpublished data).” (lines 363 – 365)

c. Who are the effector cells that cause neurodegeneration? Are they T-cells? B-cells? Microglia? Monocytes? Astrocytes? Microglia numbers are provided by the authors but despite a decline in microglial activation after normalization of IOP damage continues. Why?

Response: We have now added: “At this point, little is known regarding which subsets of T cells serve as effector cells or act as initiators of glaucomatous neurodegeneration. Involvement of multiple different effector T cell subsets in experimental autoimmune encephalomyelitis, via overlapping or distinct mechanisms, has been documented. Similar mechanisms can also be involved in glaucomatous

neurodegeneration considering that IOP elevation induced activations of all four subsets of CD4⁺ T cells in the spleens and draining LN.” “Activated CD4⁺ T cells can cause neuronal damage directly as seen in multiple sclerosis or by activating other immune cell types, such as microglia, through secreted cytokines such as IFN- γ .” (lines 376 – 392)

“d. Although one can envision how T-cells “see” HSPs when there is cell/tissue damage it is unclear how they would “see” HSPs in an intact RGC.”

Response: We have now further clarified in the Discussion section: “It has been reported that membrane-bound and extracellular HSPs, which are likely to be seen by antigen presenting cells and T cells, elicit immune responses of the adaptive or innate immune system³². Our results show that elevation of IOP upregulates membrane-bound and extracellular HSPs in the GCL, subsequently leading to immune-mediated neural damage through activating HSP-specific CD4⁺ T cells which are originally induced by microbial HSPs.” (lines 354 – 357).

Reviewers' comments:

Reviewer #1 (Remarks to the Author):

This reviewer remains unconvinced regarding the suggested relationships between the elevation of IOP and neuronal loss. The authors did not perform some of the critical experiments suggested, and instead cited studies by other scientists.

Additional comments:

Figure 1 d:

- Percentage of CD4 T cells needs to be defined relative to the parent population in the graph.
- T cell subtypes in the retina need to be gated out of total leukocytes. Since different subtypes of T cells were identified in the CLN and spleen, the same subtypes must be identified in the retina to verify exactly which CD4 T cell subset is infiltrating at the different time-points tested. The detailed characterization of the infiltrating T cell subtypes will further substantiate whether this model leads to autoimmunity.
- A graph showing the absence of CD8 T cells is missing.

Figure 4

Adoptive transfer of HSP-specific T cells or OVA T cells and RGC/AXON count must be performed in Rag mice as well, to prove that HSP specific T cells are the ones that mediate the pathology.

Figure 5

e. The scale needs to be changed. Asterisks indicating significance are missing from the ASF graph.

Supplementary 4

- a. The graph needs to be more clearly presented.
- b. Three mice per group are not sufficient for IHC.

•

Reviewer #2 (Remarks to the Author):

The manuscript raised high concerns about the roles of microbiota infection in glaucomatous pathogenesis. After revision, it has been now progressively improved, but still remains some questions unanswered.

General comments:

1. TGF-beta is not more convincing than Foxp3 as a marker for Treg. Why did the authors choose TGF-beta as the indicator of Tregs? And the discussion of the roles of those subpopulations of CD4+ cells is not sufficient.

2. Line 290:

The conclusion that HSP27- and HSP60-specific B cell responses are also elevated in patients with POAG and NTG, drawn from the titers of HSP27- and HSP60-specific IgGs titers changes, is not so persuasive. Direct evidence is needed.

3. Line 388:

Authors mentioned and cited the weakening of blood-retinal barrier several times: did they observe direct evidences proving the blood-retinal barrier was hyperpermeable in the MB injected eyes in their study, e.g. fluorescein angiograph, subretinal leakage ... ? The citation 38 seemed to be improper here as it talked more about the retardation of blood flow in glaucoma patients, providing insufficient grounds here.

4. Lines 382-392

The authors still have not answered the question why the damage is limited in retina and ON.

Minor comments:

Please correct the "x" indicating multiplication with "×" in necessity.

Figure 1d. The bar before Saline should be removed.

Reviewer #3 (Remarks to the Author):

The authors have addressed comments adequately

Reviewer #1:

“This reviewer remains unconvinced regarding the suggested relationships between the elevation of IOP and neuronal loss. The authors did not perform some of the critical experiments suggested, and instead cited studies by other scientists.”

Response: We are thankful for this reviewer’s comments as they have led to the improvement of our manuscript. However, we are confused by this comment of the reviewer and not sure which experiment it refers to. We would also like to respectfully disagree with the reviewer in that citing studies by others may present less support for coming to our conclusions. On the contrary, we believe that peer-reviewed evidence published by other groups should present even stronger support for our argument than when we would present evidence collected solely by our own laboratory. We hope the reviewer can agree with us on this.

“Figure 1 d:

- Percentage of CD4 T cells needs to be defined relative to the parent population in the graph.*
- T cell subtypes in the retina need to be gated out of total leukocytes. Since different subtypes of T cells were identified in the CLN and spleen, the same subtypes must be identified in the retina to verify exactly which CD4 T cell subset is infiltrating at the different time-points tested. The detailed characterization of the infiltrating T cell subtypes will further substantiate whether this model leads to autoimmunity.*
- A graph showing the absence of CD8 T cells is missing.”*

Response: We have now clarified that the percentage of CD4⁺ T cells was gated out of IFN- γ ⁺ cells in Fig. 1D. Indeed, we analyzed the CD4⁺ T cell subsets in the retina with flow cytometry as we did for the CLN and spleen; however, likely due to the small number of T cells that have infiltrated the retina, the only subset we were able to detect was IFN- γ ⁺ CD4⁺ cells. This has now been clarified (Page 4, lines 102 – 106, and Fig. 1 legend/Page 24, lines 760 – 762).

We are somewhat confused about what the reviewer is looking for with the request for a graph showing the absence of CD8⁺ T cells. The image would be very similar to the graph shown in the “Saline” group of Fig. 2D. We have now added “(data not shown)” in the text to clarify that such an image is not provided (Page 4, line 100).

“Figure 4. Adoptive transfer of HSP-specific T cells or OVA T cells and RGC/AXON count must be performed in Rag mice as well, to prove that HSP specific T cells are the ones that mediate the pathology.”

Response: We performed this study, but we do not quite understand the reviewer’s reasoning that performing adoptive transfer of HSP-specific T cells into *Rag1*^{-/-} mice is needed to prove that the HSP-specific T cells mediate the pathology, especially given that we have already showed the data in WT mice. Our result demonstrated that adoptive transfer of T cells isolated from HSP27-immunized mice significantly exacerbated RGC and axon loss in glaucomatous B6 mice. The immune environment in *Rag1*^{-/-} mice has little similarity to that of human patients with glaucoma, partly due to the mice’s lack of both T and B cells and low cellularity in the thymus. Nevertheless, we have performed the experiment suggested by the reviewer, and the results show that adoptive transfer of T cells of HSP27-immunized mice, but not of OVA-immunized mice, augments RGC loss in *Rag1*^{-/-} mice too. We mention this result in the Discussion section (Page 10, lines 321 – 324).

“Figure 5e. The scale needs to be changed. Asterisks indicating significance are missing from the ASF graph.”

Response: Statistical significance asterisks have now been added to the ASF group in Fig. 5e,f.

“Supplementary 4

a. The graph needs to be more clearly presented.

b. Three mice per group are not sufficient for IHC.”

Response: We have now provided better quality graphs for Supplementary Fig. 4. In addition, we performed IHC in 2 additional mice and found consistent upregulation of HSP-27 in the GCL of GF SW mice following IOP elevation, similar to that was seen in SPF mice. The data have now been updated (Supplementary Fig. 4).

Reviewer #2:

“1. TGF-beta is not more convincing than Foxp3 as a marker for Treg. Why did the authors choose TGF-beta as the indicator of Tregs? And the discussion of the roles of those subpopulations of CD4+ cells is not sufficient.”

Response: We appreciate the reviewer’s point and understand that Foxp3, being a transcription factor, is a well-defined marker for Treg while anti-TGF β may label additional CD4⁺ T cell subpopulations that exert a regulatory function, such as T_{H3} cells. Rather than trying to distinguish the involvement of Treg from other CD4⁺ T cell subpopulations (e.g. T_{H3} cells), we used cytokine profiles to define functional T cell subsets, which are known key contributors to immune diseases. The characterization is likely to be more inclusive than Foxp3 immunolabeling which may exclude other potential CD4⁺ T cell players, such as T_{H3}. We have now clarified this notion and further expanded on the discussion of the roles of the subpopulations of CD4⁺ T cells in glaucoma, as recommended by the reviewer (Page 12, lines 388 – 400).

“2. Line 290:

“The conclusion that HSP27- and HSP60-specific B cell responses are also elevated in patients with POAG and NTG, drawn from the titers of HSP27- and HSP60-specific IgGs titers changes, is not so persuasive. Direct evidence is needed.”

Response: We thank for the reviewer’s comment and have now revised the sentence to state that “the levels of HSP27- and HSP60-specific autoreactive T cells and autoantibodies are also elevated in patients with POAG and NTG.” (Page 9, lines 290 – 291).

“3. Line 388:

“Authors mentioned and cited the weakening of blood-retinal barrier several times: did they observe direct evidences proving the blood-retinal barrier was hyperpermeable in the MB injected eyes in their study, e.g. fluorescein angiograph, subretinal leakage ... ? The citation 38 seemed to be improper here as it talked more about the retardation of blood flow in glaucoma patients, providing insufficient grounds here.”

Response: We have now replaced the citation #38 with the paper by Plange et al, “Optic disc fluorescein leakage and intraocular pressure in primary open-angle glaucoma” (Curr Eye Res. 20012; 37(6):508-12) and added another citation reporting disruption of blood-retina barrier in DBA/2J mice (Ref. #39). We have further clarified: “Disruption of the blood-retina barrier has been reported in patients with POAG and in DBA/2J mice, correlating with the IOP elevation (Plange et al, 2012 Curr Eye Res; Mo JS et al 2003 J Exp Med). Although we did not detect apparent vessel leakage in microbead-induced glaucomatous mice using fluorescein angiography (data not shown), we did show T cell infiltration in MB-injected retina.” (Page 12, line 411 – 415)

“4. Lines 382-392

The authors still have not answered the question why the damage is limited in retina and ON.”

Response: We have now further clarified that glaucomatous neural damage is limited to the retina and optic nerve because elevated IOP selectively affects these areas and causes inflammation (Page 11, lines 372 – 374). We apologize for not making this clear in our last submission.

“Please correct the “x” indicating multiplication with “×” in necessity.”

Response: We have now gone through the entire manuscript and replaced “x” indicating multiplication with “×”.

“Figure 1d. The bar before Saline should be removed.”

Response: This has now been corrected.

REVIEWERS' COMMENTS:

Reviewer #1 (Remarks to the Author):

The authors addressed most of the comments raised by the two referees. This reviewer has no further comments except for the original ones, that were mostly addressed.

Reviewer #2 (Remarks to the Author):

General comment:

Considering the CD4 positive events might only count up to 200-300 each sample, the subsequent analysis shown in suppl. Fig. 2b & c might be flimsy. Besides, the proposed discussion about the subgroups of the CD4 positive cells did not refer to the general role in the immune system as the tedious citation (Line 388-400) here, but to the role to be interpreted by the authors in the pathogenesis in this glaucoma model.

No other comments as the former ones are answered.

Minor comment:

Figure 4: Script "b" missing.

Responses to reviewers' comments:

“Considering the CD4 positive events might only count up to 200-300 each sample, the subsequent analysis shown in suppl. Fig. 2b & c might be flimsy. Besides, the proposed discussion about the subgroups of the CD4 positive cells did not refer to the general role in the immune system as the tedious citation (Line 388-400) here, but to the role to be interpreted by the authors in the pathogenesis in this glaucoma model.”

Response: Counts of CD4+ T cells up to 200-300 each sample as referred by the reviewer were taken from the retina, while suppl. Fig. 2 analyzes systemic immune responses by counting cells from the cervical lymph nodes and spleen, where there are plenty of CD4+ T cells. We have now further clarified this point in suppl. Fig. 2 legend. In addition, we have now also highlighted the paragraph referring to the general role of CD4+ T cells in the immune system, but not that relating specifically to the pathogenesis of glaucoma model (lines 394 - 399).

“Figure 4: Script “b” missing.”

Response: This has been corrected.